# Single cell analysis of human foetal liver captures the transcriptional profile of hepatobiliary hybrid progenitors

Joe M. Segal [1,8], Deniz Kent[1,8], Daniel J. Wesche[2,3], Soon Seng Ng [1], Maria Serra[1], Bénédicte Oulès [1], Gozde Kar[4], Guy Emerton[4], Samuel J.I. Blackford [1], Spyros Darmanis[5], Rosa Miquel[1], Tu Vinh Luong[1], Ryo Yamamoto[2], Andrew Bonham[2], Wayel Jassem[6], Nigel Heaton[6], Alessandra Vigilante[1], Aileen King[7], Rocio Sancho [1], Sarah Teichmann [4], Stephen R. Quake[5,9], Hiromitsu Nakauchi[2,9] & S. Tamir Rashid[1,2,9]

The liver parenchyma is composed of hepatocytes and bile duct epithelial cells (BECs). Controversy exists regarding the cellular origin of human liver parenchymal tissue generation during embryonic development, homeostasis or repair. Here we report the existence of a hepatobiliary hybrid progenitor (HHyP) population in human foetal liver using single-cell RNA sequencing. HHyPs are anatomically restricted to the ductal plate of foetal liver and maintain a transcriptional profile distinct from foetal hepatocytes, mature hepatocytes and mature BECs. In addition, molecular heterogeneity within the EpCAM$^+$ population of freshly isolated foetal and adult human liver identifies diverse gene expression signatures of hepatic and biliary lineage potential. Finally, we FACS isolate foetal HHyPs and confirm their hybrid progenitor phenotype in vivo. Our study suggests that hepatobiliary progenitor cells previously identified in mice also exist in humans, and can be distinguished from other parenchymal populations, including mature BECs, by distinct gene expression profiles.

[1] Centre for Stem Cells and Regenerative Medicine & Institute for Liver Studies, King's College London, London WC2R 2LS, UK. [2] Institute for Stem Cell Biology and Regenerative Medicine, Stanford University School of Medicine, Stanford, CA 94305, USA. [3] Department of Microbiology and Immunology, Stanford University School of Medicine, Stanford, CA 94304, USA. [4] Wellcome Trust Sanger Institute, Hinxton CB10 1SA, UK. [5] School of Engineering, Stanford University, Stanford 94350 CA, USA. [6] Institute of Liver Studies, Kings College Hospital, London SE4 9RS, UK. [7] Department of Diabetes, King's College London, London SE1 1UL, UK. [8] These authors contributed equally: Joe M. Segal, Deniz Kent. [9] These authors jointly supervised this work: Stephen R. Quake, Hiromitsu Nakauchi, S. Tamir Rashid. Correspondence and requests for materials should be addressed to J.M.S. (email: joe.segal@kcl.ac.uk) or to S.T.R. (email: tamir.rashid@kcl.ac.uk)

In rodents both hepatocytes and biliary epithelial cells (BECs) are derived from a common bi-potent hepatoblast population during liver development[1]. In adult mice, conflicting evidence exists regarding the presence of a distinct bi-potent progenitor capable of regenerating both hepatocytes and BECs. The regenerative potential of the rodent liver has been attributed to hepatocytes[2], BECs[3,4], biliary-like progenitor cells or 'oval cells' arising in the ductal region[5,6], stem cells located around the central vein[7] and hepatocyte or cholangiocyte de-differentiation into a hybrid bi-potent progenitor[8,9]. In comparison, the mechanisms of human liver regeneration are poorly characterised. It has been proposed that EpCAM$^+$ human liver stem/progenitor cells reside in the ductal plate (DP) during foetal liver development. After birth these cells localise to the Canals of Hering, where upon severe chronic liver injury they become reactivated forming what is pathologically described as ductular reactions[10,11]. Despite these findings, the existence of a bi-potent human liver 'progenitor' cell remains unclear. This issue is in part due to a substantial overlap in markers between potential progenitor populations, hepatic precursors and mature BECs[3,12,13], challenging the field to define the true transcriptional nature of a bi-potent progenitor phenotype that can be replicated for clinical use. Several recent studies have captured a bi-potent progenitor-like state via small molecule-reprogramming of primary hepatocytes, capable of in vitro hepatic and biliary maturation, imitating a process that has been observed during chronic mouse liver injury[8,14–16]. Despite several well-established phenotypic criteria for liver progenitor cells, no benchmark exits that truly distinguishes them from other human hepatic and biliary cells. To facilitate the in vitro development of cell-based therapies for treating liver disease, it is critical to precisely define a liver progenitor cell that accurately captures the developmental origin of human liver parenchyma.

In this study we utilise single-cell RNA sequencing (scRNA-seq) to interrogate the transcriptome of human foetal and adult liver at single-cell resolution. In recent years scRNA-seq has helped identify unreported cell types within populations previously defined as homogenous[17–20]. Here, we report the transcriptional signature of distinct hepatic cell types in foetal and adult human liver, including a foetal hepatobiliary hybrid progenitor (HHyP) population. Capturing a human hepatic progenitor state in utero provides unparalleled and unexplored insight into the true mechanisms of human liver development. We identify a gene expression profile that can distinguish between foetal HHyPs, foetal hepatocytes and mature BECs. We further identify HHyP-like cells maintained in uninjured adult primary liver tissue. Finally, we FACS sorted HHyPs from freshly isolated human foetal liver and show evidence of hepatic and biliary phenotypes in vivo. Our in depth profiling of previously undefined HHyPs finally provides an accurate template for the human liver progenitor phenotype that will be a valuable roadmap for translating ex vivo hepatic progenitor studies into successful cell-based liver disease therapies.

## Results

**EpCAM$^+$ cell heterogeneity in human foetal and adult liver by scRNA-seq.** To capture the cellular heterogeneity of human liver during development, we combined a FACS strategy with scRNA-seq. We first sorted by negative selection of red blood cells (CD235a) and immune cells (CD45), and positively selected for EpCAM and NCAM to enrich for potential human liver progenitors[10,21] (Supplementary Fig. 1). To investigate how foetal human liver populations progress into adult liver we isolated and sequenced EpCAM$^+$ (biliary cells) and EpCAM$^−$/ASGPR1$^+$ (mature hepatocytes) cells from fresh, uninjured adult tissue

(Supplementary Fig. 1). In total, 1224 cells were sequenced from human foetal and adult livers. Following stringent quality control (qc), 741 cells were retained for downstream analyses (Supplementary Fig. 1)[22]. Sample counts were normalised as transcripts per million (TPM).

To define different populations captured by our FACS strategy, we employed t-Distributed Stochastic Neighbour Embedding (t-SNE) on high-variance genes. We then measured differential gene expression to phenotypically characterise the different cell groups (Fig. 1a, b). We identify several distinct ALB$^+$ cell populations in both the adult and foetal liver single cell analysis. As expected by sorting adult human liver cells by EpCAM expression, nearly all ASGPR1$^+$ cells are identified as ALB$^+$/ASGR1$^+$/AFP$^−$ mature hepatocytes (Supplementary Data 1). EpCAM$^+$ adult cells express progenitor/BEC markers KRT19 and SOX9, but interestingly most are highly ALB$^+$ as well (Fig. 1b, c, Supplementary Data 2). In foetal liver, two distinct ALB$^+$ expressing populations and several non-hepatic populations transcriptionally resembling stromal, mesothelial and erythroblast cell types were found (Supplementary Fig. 2). Within the ALB$^+$ populations we identified a foetal hepatocyte population expressing hepatoblast markers AFP and DLK1, but negative for traditional biliary markers (Fig. 1b, c, Supplementary Data 3). Exclusively within the CD235a$^−$/CD45$^−$/EpCAM$^+$/NCAM$^+$ population of human foetal liver we identify a cluster of cells that closely resembles EpCAM$^+$ biliary cells identified in adult liver, expressing both biliary and hepatic markers which we label here as HHyP cells (Fig. 1b, c). Foetal HHyP cells expressed hepatic genes ALB, APOE, TF and HNF4A, but were also positive for BEC markers KRT19, SOX9 and CD24 (Fig. 1c, d, Supplementary Data 4). Having expected to isolate mature BECs from adult EpCAM$^+$ cells we looked for expression of mature BEC markers recently identified in a comprehensive transcriptomic map of adult human liver[23]. We identify a small ALB$^−$ population enriched for mature BEC markers TFF1 and TFF2. This BEC population is transcriptionally distinct from HHyPs which co-express hepatic markers, liver progenitor markers and mature biliary markers (Fig. 1c, d, Supplementary Data 5). Whilst potential BECs in our study share many genes with HHyPs from both foetal and adult human liver, they express a subset of genes enriched in mature human BECs captured by scRNA-seq[23] (Supplementary Fig. 3). Whilst potential BECs in our study share many genes with HHyPs from both foetal and adult human liver, we have identified differences in gene expression that can distinguish BECs from HHyPs (Supplementary Fig. 3)[23]. After applying phenotypic labelling to t-SNE analysis we see contribution from multiple donors for each cell type, including rarer populations, demonstrating the robustness of our data set (Fig. 1e, f).

Populations of hybrid bi-potent progenitors have previously been characterised in mice after chronic liver injury[8]. We, therefore, compared the transcription profiles of human ALB$^+$/KRT19$^+$/KRT7$^+$ HHyPs and ALB$^−$/KRT19$^+$/KRT7$^+$ BECs from our data set with mouse hepatocyte-derived proliferative ducts (HepPD) and biliary-derived proliferative ducts (BilPD) from Tarlow et al.[8]. Relative to their respective biliary populations, human foetal and adult HHyPs have similar expression patterns to HepPDs, identified in mice as having bi-potent characteristics of liver progenitor cells[8]. Genes, including AHSG, RBP4, SFRP5 and MCAM are enriched in both human HHyPs and mouse HePDs, while mature BEC markers KRT7, MUC1, TSPAN8 and TFF2 are downregulated in both (Supplementary Fig. 4). Thus, it is likely that our single-cell strategy has captured the existence of a hepatobiliary hybrid progenitor population that is transcriptionally distinct from mature BECs and other hepatic cell populations, but similar to hybrid progenitor cells identified in mice after chronic liver injury.

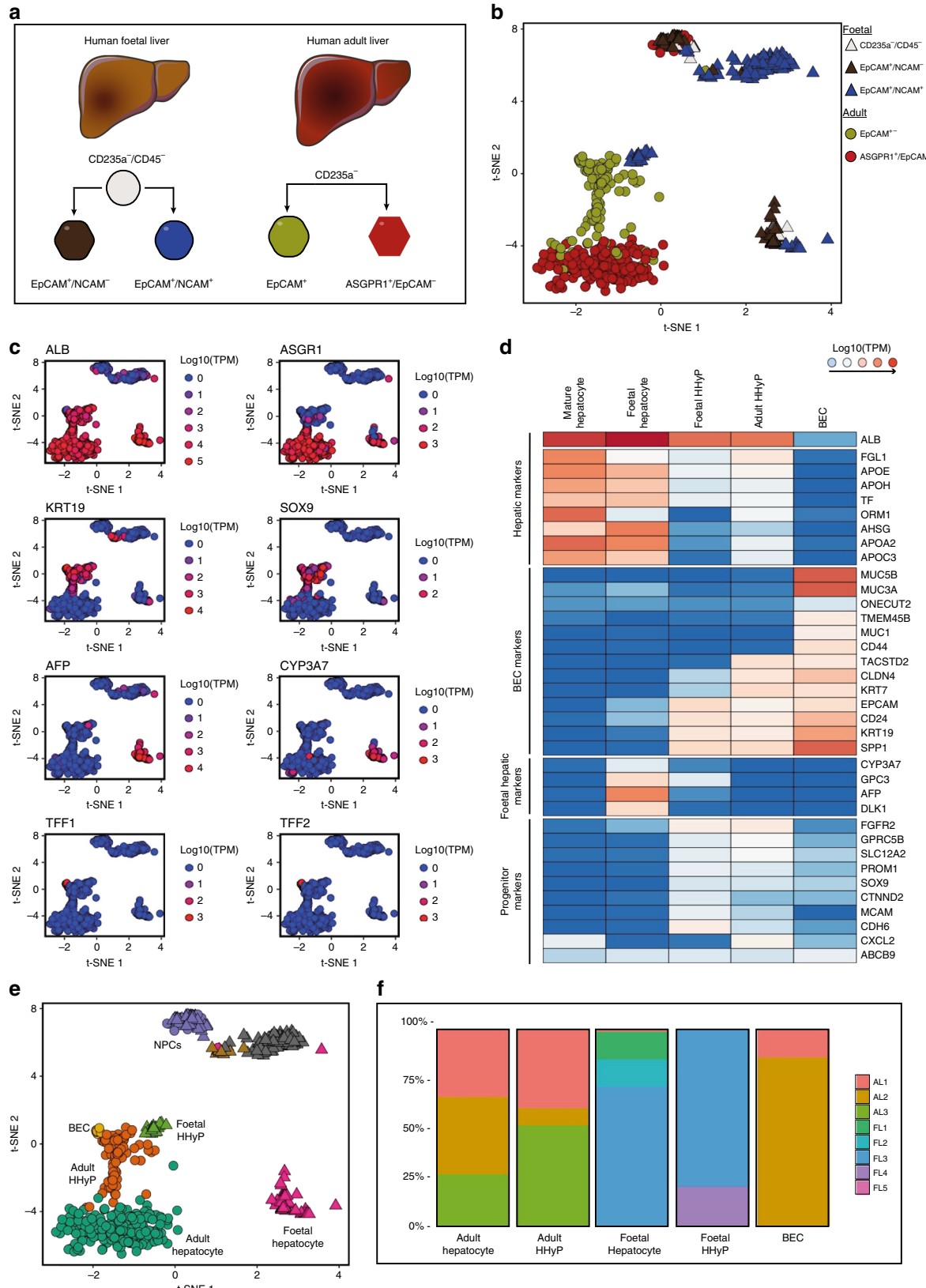

**Identification of a distinct foetal HHyP transcriptional phenotype**. Further in-depth characterisation of the markers defining different liver populations is required to fully understand their role in development and liver regeneration. Foetal hepatocytes and HHyPs demonstrate clear transcriptional distinction. Despite sharing *ALB* gene expression, foetal hepatocytes express none of the traditional progenitor/BEC markers enriched in HHyPs, including *KRT7, SPP1, STAT1, SOX9* and *HNF1B*. (Fig. 2). Upon t-SNE analysis performed only on *ALB*+ cells from our study, K-means clustering indicates that adult HHyPs are more closely

**Fig. 1** ScRNA-Seq analysis of foetal and adult human liver. **a** Overview of foetal and adult liver FACS strategy. **b** 2D t-SNE visualisation of single cells isolated from foetal and adult human liver coloured by FACS gating population, shaped by tissue source. **c** Transcript expression of selected markers overlaid on the 2D t-SNE space of human liver scRNA-seq analysis. Expression is Log10(TPM). **d** Heat maps of selected gene expression in mature hepatic, foetal hepatic, hybrid hepatobiliary progenitor (HHyP) and mature cholangiocyte cell populations. Gene expression in Log10(TPM). Mean gene expression of cells in each cluster is plotted, HHyP population = 138 cells, mature biliary epithelial cell (BEC) = 9 cells, mature hepatocytes = 226 cells, foetal hepatocytes = 82 cells. **e** 2D t-SNE visualisation of single cells isolated from foetal and adult human liver coloured by cell type. Phenotypic labelling based on transcriptional analysis. **f** Proportions of tissue sample contributions from adult liver (AL) and foetal Liver (FL) in each phenotypically labelled cell type as a percentage of the total population. t-SNE t-distributed stochastic neighbour embedding, TPM transcripts per million, FACS fluorescence-activated cell sorting

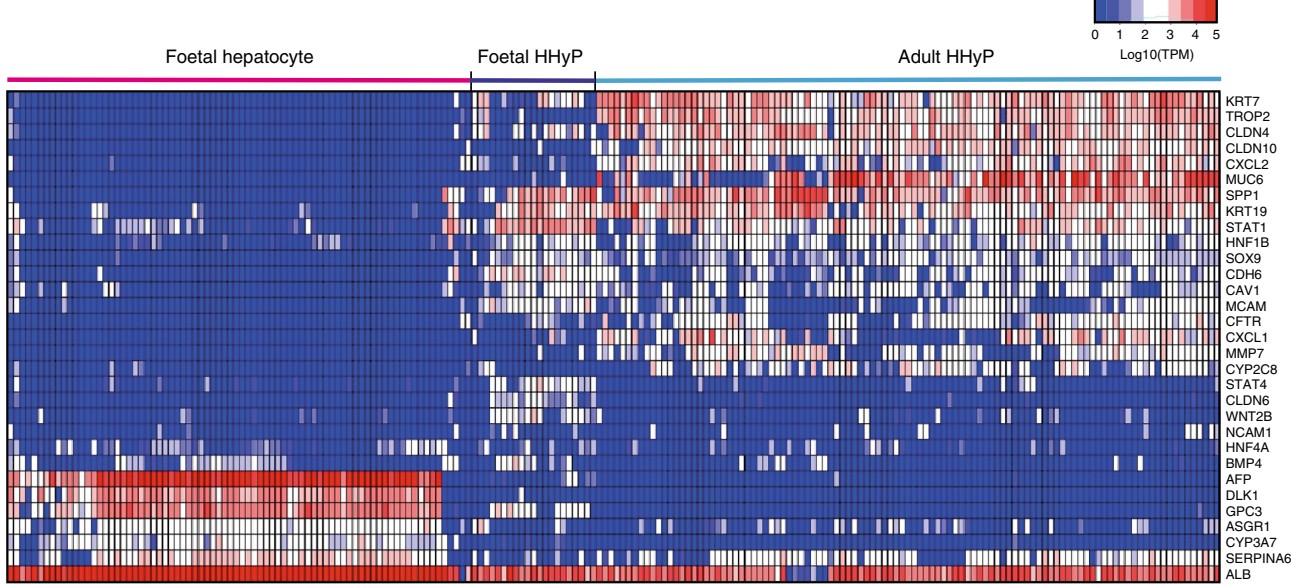

**Fig. 2** Comparison of foetal hepatic and HHyP scRNA-seq populations. Heat map of selected gene expression in foetal hepatic, foetal hybrid hepatobiliary progenitor (HHyP) and adult HHyP cells. Gene expression in Log10(TPM). TPM transcripts per million

associated with adult mature BECs then foetal HHyPs (Fig. 3a). Studies have suggested uninjured adult liver would not contain progenitor-like cells with a hybrid hepatobiliary phenotype, yet our scRNA-seq analysis on uninjured human adult liver demonstrates HHyPs are both present and negative for many recently identified mature human BEC markers[23]. However, despite the high transcriptional similarity between foetal and adult HHyPs there are some key differences in gene expression (Fig. 3b). Gene set enrichment analysis (GSEA) on foetal HHyP genes enriched over adult identifies 'Stem cell proliferation', 'Developmental cell growth', 'Homophilic cell adhesion via plasma membrane adhesion molecules' and association with 'Extracellular matrix component', suggesting a phenotype associated with progenitor/stem cell-like function and important interactions with the niche environment (Fig. 3c). Genes, including *TACSTD2 (TROP-2)*, *CLDN4*, *CLDN10* and *KRT7* are enriched in adult HHyPs compared to the foetal HHyP population (Fig. 3d). In contrast, expression of *CLDN6* and the transcription factor *STAT4* are exclusively detected in foetal HHyPs. *MCAM*, *CDH6* and *STAT1* are also enriched in foetal HHyPs over adult (Fig. 3d). We observed several other interesting gene expression patterns, including *GPC3* expressed exclusively in foetal parenchymal populations, *CXCL2* expressed exclusively in adult parenchymal populations and *MUC1* restricted to BECs (Fig. 3d). To further understand how foetal HHyPs related to the current understanding of liver progenitor cells in the field, we looked at the transcriptome of a recent study for human primary hepatocyte-derived liver progenitor-like cells[16]. We identified that

many of the top genes enriched in foetal HHyPs are enriched during the transition process from primary hepatocytes to progenitor-like cells, including *MCAM*, *ANXA2*, *ANXA4*, *BICC1*, *SPIN1*, *TNFRSF12A*, *STAT1* and *ABCC3* (Fig. 3e). This suggests that in vitro reprogramming techniques to create progenitor-like cells from mature hepatocytes are moving towards a foetal HHyP-like phenotype.

**TROP-2 expression is restricted to biliary committed cells**. To validate the transcriptional signature of HHyP cells we next employed RNA in situ hybridisation (RNA-i*sh*) and characterised their spatio-temporal regulation in primary human foetal liver tissue. Expression of the HHyP markers *CDH6*, *STAT1*, *CD24*, *FGFR2*, *DCDC2* and *CTNND2* that we identified are restricted to the ductal plate (DP) (a layer of cells surrounding the portal tract) in second trimester human foetal liver[24] (Fig. 4a, Supplementary Fig. 5). At this stage *ALB* is expressed highly in the parenchyma, but absent from intrahepatic bile ducts (BDs) (Supplementary Fig. 5). We confirmed the hepatic phenotype of DP cells by co-expression of the DP marker *STAT1* with the hepatic marker *HNF4A* (Supplementary Fig. 5). We, additionally, observed that CDH6 and STAT1 co-localised with the previously reported human hepatic stem cell marker CLDN3 and biliary marker SOX9 in the DP at the protein level (Fig. 4b, Supplementary Fig. 5)[5,10]. Importantly, these markers were also expressed within foetal BDs alongside CK19, a classical DP marker (Fig. 4c)[24]. These results highlight a significant challenge to the field, distinguishing between potential bi-potent liver progenitors and biliary committed cells

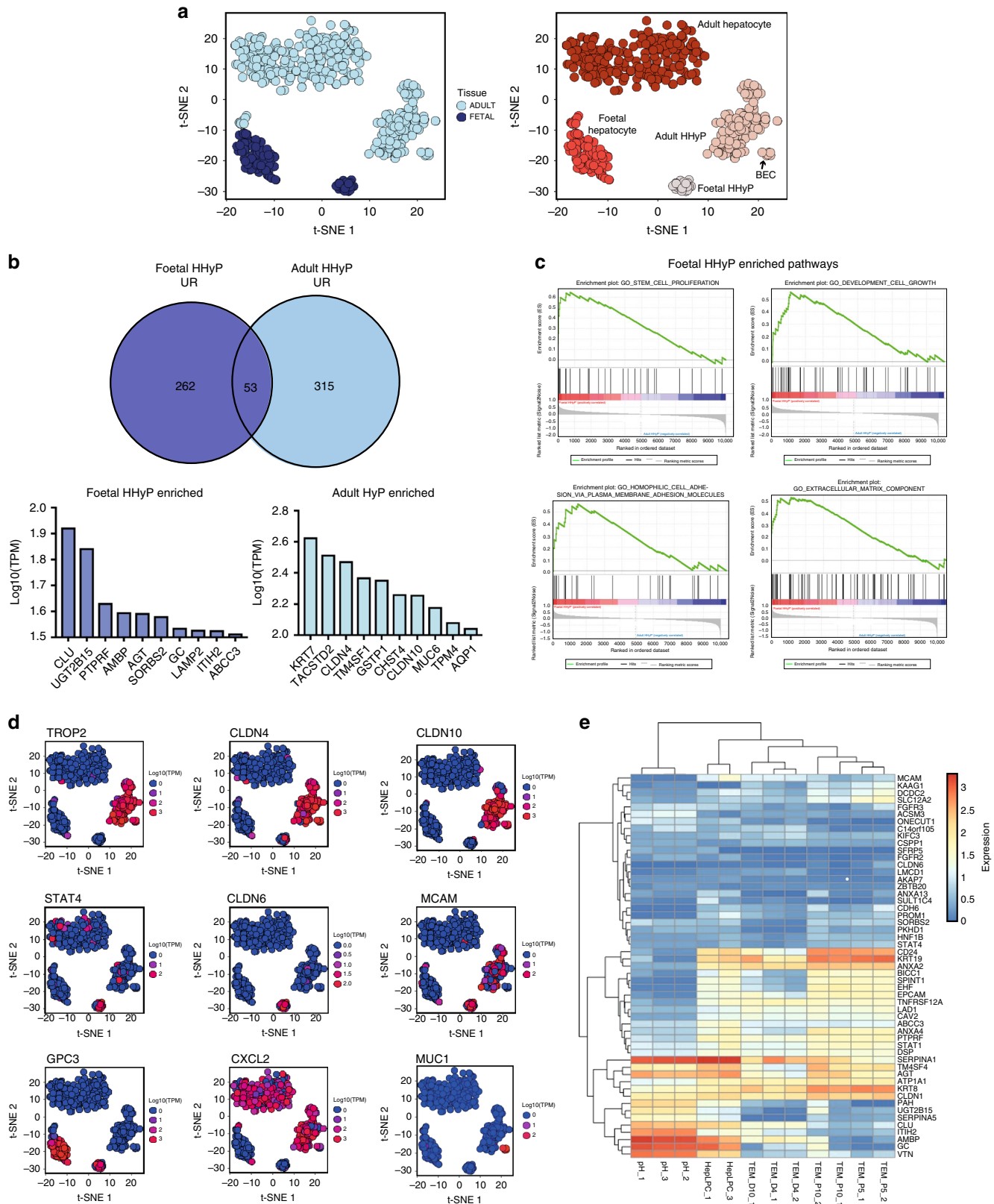

which share several markers. Our scRNA-seq analysis identified *TROP-2* expression to be restricted to adult progenitor/BEC cells and absent from foetal HHyPs. Furthermore in Tarlow et al.[8], TROP-2 is expressed in bilPDs (biliary progenitors) but not bipotent hepPDs. We, therefore, investigated *TROP-2* spatial

regulation in human foetal liver, as compared to progenitor markers *CDH6* and *STAT1*. We observed by RNA-*ish* that while *CDH6* and *STAT1* expression were observed in BDs and the DP progenitor zone, *TROP-2* expression was anatomically restricted to BDs (Fig. 4d). These findings suggest that *TROP-2* is up regulated

**Fig. 3** Comparison of *ALB*+ cells in human liver scRNA-seq populations. **a** 2D t-SNE visualisation of *ALB*+ cells isolated from foetal and adult human liver coloured by tissue type (left panel) and K-means cluster (right panel). Phenotypic labelling based on transcriptional analysis. **b** Comparison of foetal and adult hepatobiliary hybrid progenitors (HHyP) significantly enriched genes (FC 1.1, *p*-val < 0.05 with student *t* test) by venn diagram with top ten highly expressed genes in foetal (left) and adult (right) HHyPs. **c** Gene set enrichment analysis (GSEA) of foetal vs. adult HHyPs for Gene ontology (GO) terms 'Stem cell proliferation', 'Developmental cell growth', 'Homophilic cell adhesion via plasma membrane adhesion molecules' and 'Extracellular matrix component'. **d** Transcript expression of selected markers overlaid on the 2D t-SNE space of human liver scRNA-seq analysis for adult HHyP-restricted genes (top), foetal HHyP-restricted genes (middle) and other cell type-specific expression patterns (bottom). Expression is Log10(TPM). **e** Heatmap of top foetal HHyP up regulated genes in publicly available sequencing data from human hepatic liver progenitor cells (hepLPCs-Heps, GSE105019) converted from primary hepatocytes[16]. Heatmap shows expression in primary hepatocytes (pH) and human primary hepatocytes converted into liver progenitor-like cells (HepLPCs) at different stages in transition and expansion medium (TEM). The colour bar indicates gene expression in log10 scale. t-SNE t-distributed stochastic neighbour embedding, TPM transcripts per million, FC fold change

during biliary lineage commitment. Therefore *TROP-2* is a key marker to distinguish human foetal liver hybrid progenitors from *TROP-2*+ committed BECs present in BDs.

We next investigated whether these in situ findings translated to ex vivo lineage commitment. Previous studies have isolated and expanded progenitor-like cells in vitro from the EpCAM+ population of human liver in 3-dimensional (3D) culture systems[10,21]. We, therefore, isolated EpCAM+ cells from human foetal liver by MACs column purification and generated 3D organoids in matrigel suspension (Fig. 4e)[21]. As expected, organoids grown in liver expansion media were positive for EpCAM, SOX9, CDH6 and HNF4A, suggesting they retained a hybrid hepatobiliary phenotype in these conditions (Fig. 4e). To trigger lineage commitment, EpCAM+ organoids were transferred to either hepatic or biliary differentiation (BD) media[13,25–28]. Consistent with our scRNA-Seq data and in situ staining, organoids became positive for TROP-2 and CK7 upon transfer to a BD media, whereas ALB and HNF4A were not expressed. In contrast, organoids transferred to hepatic differentiation (HD) media expressed ALB, while CDH6, TROP-2 and CK7 were lost in ALB/HNF4A expressing structures (Fig. 4f). These findings suggest the expression signature of foetal HHyPs from our scRNA-seq dataset can be utilised to distinguish the profile of a human liver progenitor from biliary and hepatic committed cells ex vivo.

**Adult HHyPs exhibit gene signatures of both hepatic and biliary lineage.** Our findings identified distinct human foetal and adult HHyP populations with key differences in gene expression. To further investigate heterogeneity within the foetal and adult HHyP populations, we examined their lineage potential using the R package MONOCLE[29]. Populations were clustered into 5 'pseudo states' in 2D PCA space (Supplementary Fig. 6). Pseudo state 1 contained all foetal HHyPs, and a sub-population of adult HHyPs (blue), enriched markers *DCDC2, CDH6, STAT1* and *ANXA13* (Supplementary Fig. 6). Interestingly, we observed distinct adult HHyP clusters enriched for either biliary (Pseudo state 3) or hepatic (Pseudo state 4) lineage markers. Pseudo state 3 revealed an enrichment of ductal marker genes *KRT7, KRT23* and *TROP-2*, while hepatic transcription factors *ATF5, MLXIPL* and *CREB3L3* were enriched in pseudo state 4 (Supplementary Fig. 6).

We next looked at the spatial expression of enriched pseudo state 1 markers *CDH6, DCDC2* and *ANXA13*, alongside *STAT1* in adult uninjured liver by in situ hybridisation (Supplementary Fig. 6). Intriguingly, all markers were expressed in both BDs and at the limiting plate, an embryonic remnant of the DP surrounding the portal mesenchyme, suggesting these cells may be a population of intrahepatic duct residing cells distinct from ALB−, TFF1+/TFF2+/MUC1+ BECs identified here and in other studies[23]. This is further supported by our observation that *TROP-2* expression is restricted to foetal liver BDs and not HHyPs localised to the DP (Fig. 4d). TROP-2 has previously been

identified in mouse liver injury as a marker that distinguishes cells from BECs[30], and only expressed in human cancers in the liver[31]. Therefore, it was important to confirm that *TROP-2* is highly expressed in the BDs of uninjured adult human liver by in situ (Supplementary Fig. 6).

**FACS isolation and in vivo transplantation of human foetal liver HHyPs.** We next looked to assess the intrinsic hepatobiliary lineage potential of distinct foetal human liver populations in vivo. We used FACS to isolate distinct foetal hepatic cell populations based on their differential surface marker expression, and transplanted each population individually underneath the renal capsule of 10-week-old NOD scid gamma (NSG) immunodeficient mice (Fig. 5a). This approach has previously been used to validate the differentiation potential of mouse and human stem cell populations[32–38]. We FACS sorted a number of populations to investigate their respective in vivo differentiation potential. We isolated CD235a−/CD45−/EpCAM+/NCAM+/MCAM+ HHyP cells, based on our scRNA-seq analysis of the HHyP phenotype (Fig. 5b). As a control we sorted for CD235a−/CD45−/EpCAM+/NCAM−/MCAM− cells to assess the in vivo behaviour of non-HHyP EpCAM+ cell types relative to potential HHyPs (Fig. 5c). The xenografts were analysed after 4 weeks of development within the kidney capsule, and were subsequently assessed by H&E and immunofluorescence (IF) staining to determine their expansion capabilities and lineage potential. After 4 weeks, only the HHyP cell population produced clear expanded regions, despite matched cells numbers transplanted between HHyPs and EpCAM+ only populations. Non-transplanted kidney from matched samples contained no such explant regions (Fig. 5d).

We investigated the presence of hybrid progenitor cells within the foetal HHyP explant by IF staining of hepatobiliary markers identified in our scRNA-seq analysis and validated ex vivo. IF staining of human-specific ALB and the BEC/progenitor marker CK19 within foetal HHyP explant sections revealed the presence of ALB+/CK19+ cells, as well as cells expressing only ALB (Fig. 5e). ALB staining was negative in matched mouse adult liver control, confirming the presence of human cells in the explant. We next looked at markers of hepatic and biliary commitment by IF to assess lineage potential of the FACS sorted foetal HHyP population. The hepatic markers fumarylacetoacetate hydrolase (FAH) and HNF4a are expressed widely across the explant region of HHyP cells confirming the hepatic potential of foetal HHyPs (Fig. 5f). To confirm biliary lineage potential we co-stained HHyP explants for ALB and TROP-2, a marker we identified as negative in foetal HHyPs and expressed in mature BECs by scRNA-seq analysis. We identify ALB−/TROP-2+ duct-like structures suggesting that the HHyPs population is also capable of producing mature BECs (Fig. 5e). Finally, we also isolated CD235a−/CD45−/EpCAM+/NCAM+/TROP2− cells to enrich for HHyPs over biliary lineage committed cells (Fig. 6a, b). As

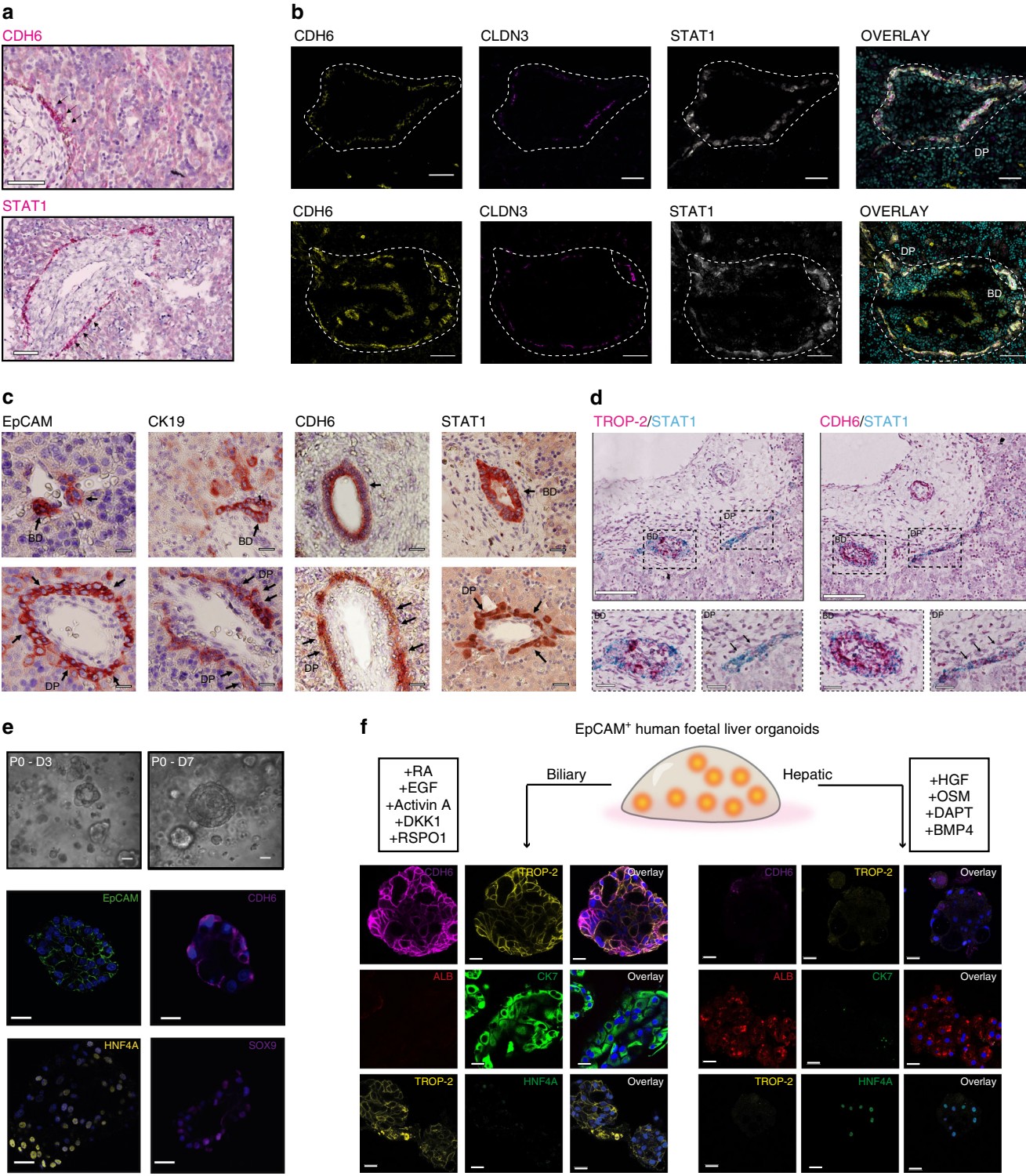

**Fig. 4** TROP2⁻ foetal HHyPs are restricted to the ductal plate of foetal liver. **a** RNA-*ISH* for *CDH6* and *STAT1* on human second trimester (15–21 pcw) foetal liver ductal plate (DP) and bile duct (BD) regions. Scale bars represent 50 μm. **b** Immunofluorescence (IF) staining of CDH6 (yellow), CLDN3 (magenta) and STAT1 (grey) co-expression in human foetal liver slides. Slides counterstained in DAPI (cyan). DP and BD structures outlined in white. Scale bars represent 25 μm. **c** Immunohistochemistry (IHC) of EpCAM, CK19, CDH6 and STAT1 in BD and DP regions of human foetal liver. Scale bars represent 50 μm. **d** Duplex RNA-ISH for *CDH6*(red)/*STAT1*(blue) and *TROP-2*(red)/*STAT1*(blue) in foetal liver BD and DP structures. Scale bars represent 50 μm. Scale bars of zoomed in region represent 25 μm. **e** Phase contrast imaging and IF of foetal intra-hepatic organoids (f-IHOs) derived from EpCAM enriched foetal liver cells in liver expansion (LE) media. All structures are counterstained with DAPI (blue). All staining performed between passages 3 and 5. **f** Schematic and IF staining of f-IHOs cultured for 7 days in hepatic differentiation (HD media) and biliary differentiation (BD media). All staining performed between passages 3 and 5. Images representative of n = 3 foetal liver differentiation experiments

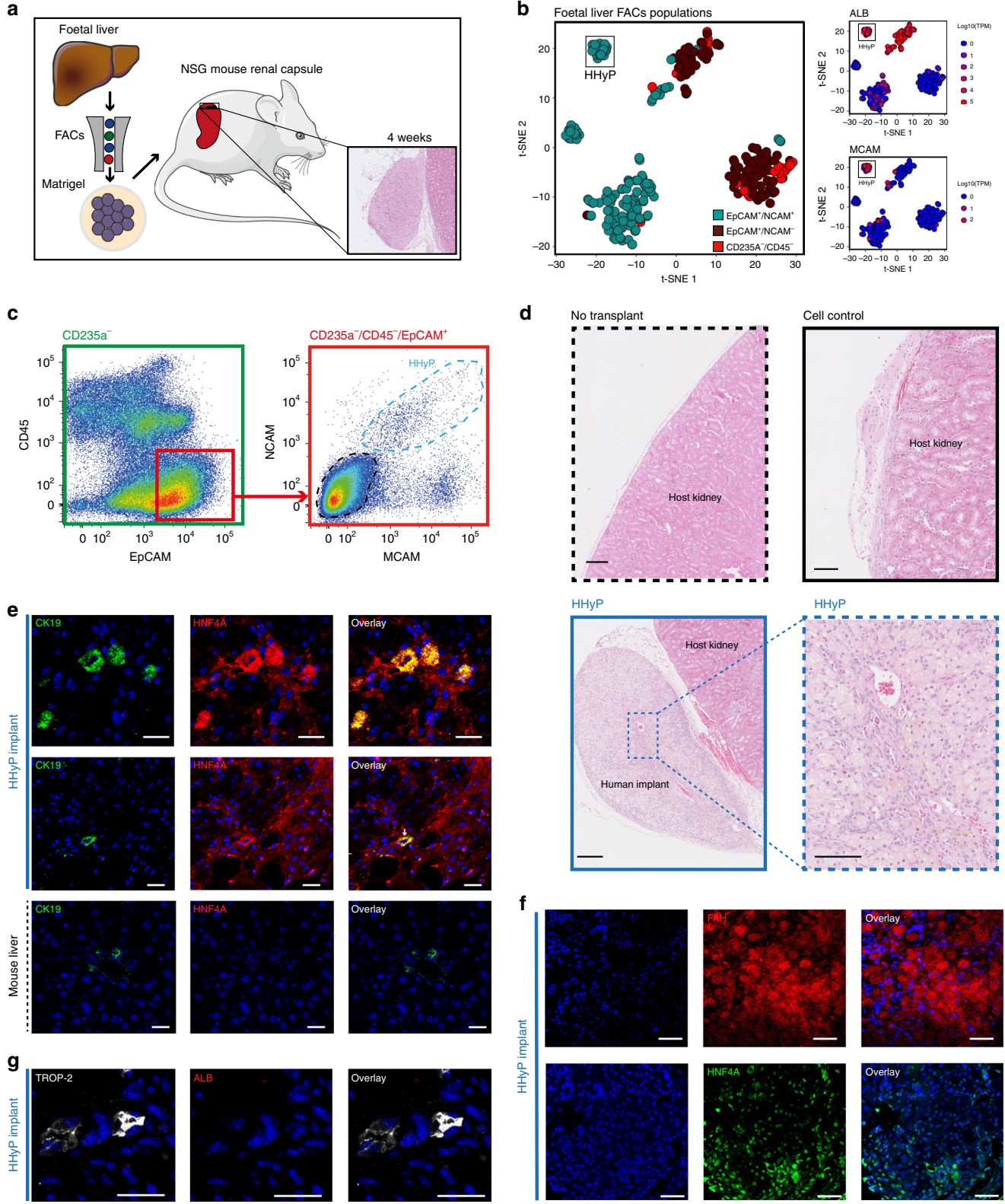

expected few cells were captured from foetal CD235a⁻/CD45⁻/EpCAM⁺/NCAM⁻/TROP2⁺ population, likely due to their restriction to forming BDs, thus proving difficult to isolate (Fig. 6c). Again only HHyP cell expansion was observed upon renal capsule transplantation over matched control cells (EpCAM⁻/NCAM⁻/TROP2⁻ cells) or BECs (Fig. 6d). We further show

that these cells express FAH, confirming their hepatic lineage potential upon in vivo implantation (Fig. 6e). Collectively, these results demonstrate the hepatobiliary hybrid phenotype of foetal HHyPs identified in our scRNA-seq analysis, as captured previously in mouse chronic liver injury studies and small-molecule reprogramming of primary human hepatocytes.

**Fig. 5** In vivo lineage potential of human foetal HHyPs. **a** Experimental strategy for isolation and in vivo characterisation of foetal hepatobiliary hybrid progenitors (HHyPs) by transplantation beneath the renal capsules of immunodeficient NOD scid gamma (NSG) mice. **b** 2D t-SNE visualisation of single cells isolated from foetal human liver coloured by FACS gating population (left panel) and transcript expression of ALB and MCAM overlaid on the 2D t-SNE space of human foetal liver scRNA-seq analysis (right panel). Expression is Log10(TPM). **c** Gating scheme for the isolation of distinct foetal human liver populations based on expression of CD235a, CD45, EpCAM, NCAM and MCAM. **d** Hematoxylin and eosin (H&E) staining in tissue cross-sections of implant regions 4 weeks post renal capsule transplantation of human foetal liver FACS populations. Scale bars represent 100 μm. **e** Immunofluorescence (IF) co-staining of CK19 (green) and ALB (red) in implant regions of HHyPs post 4 weeks transplantation and matched control mouse adult liver. Slides counterstained in DAPI (cyan). Scale bar represents 25 μm. **f** IF staining of FAH (red) and HNF4A (green) in implant region of HHyPs 4 weeks post transplantation. Slides counterstained in DAPI (cyan). Scale bars represent 25 μm. **g** IF co-staining of TROP-2 (white) and ALB (red) in implant region of HHyPs 4 weeks post transplantation. Slides counterstained in DAPI (cyan). Scale bars represent 25 μm. t-SNE t-distributed stochastic neighbour embedding, FACS fluorescence-activated cell sorting, TPM transcripts per million

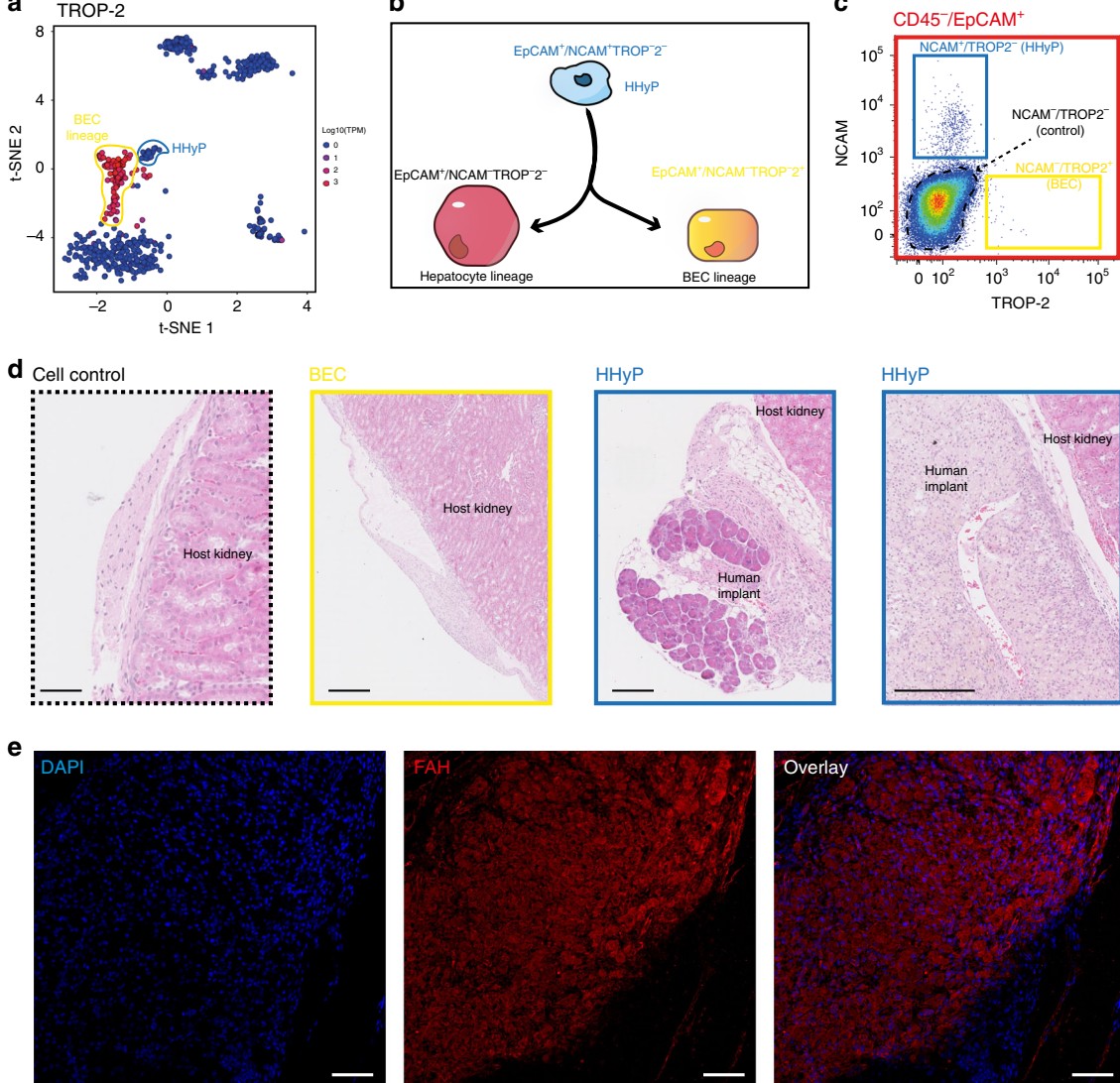

**Fig. 6** Expansion of EpCAM+/NCAM+/TROP2− human foetal liver HHyPs in the renal capsule. **a** 2D t-SNE visualisation of single cells isolated from foetal and adult human liver coloured by TROP-2 gene expression. Expression is Log10(TPM). **b** Schematic of hepatic and BEC lineage restriction of HHyPs based on NCAM/TROP-2 expression. **c** Gating scheme for the isolation of distinct foetal human liver populations based on expression of CD235a, CD45, EpCAM, NCAM and TROP2. **d** Hematoxylin and eosin (H&E) staining in tissue cross-sections of explant region for 4-week post renal capsule transplantation. Scale bars represent 250 μm. **e** Immunofluorescence (IF) staining of FAH (red) in explant region of CD235a−/CD45−/EpCAM+/NCAM+/TROP2− HHyPs 4-week post transplantation. Slides counterstained in DAPI (cyan). Scale bars represent 75 μm. t-SNE t-distributed stochastic neighbour embedding, TPM transcripts per million

Our FACS and scRNA-seq strategy has captured the transcriptional profile of a HHyP population that arises during human foetal liver development. We utilised this profile to clarify the strong overlap in markers between potential progenitor populations and mature BECs and define a gene signature capable of truly distinguishing liver progenitor cells from mature BEC populations. Integrating our foetal data set with recent scRNA-seq and bulk analysis of different hepatic populations, we identify marker sets that exclusively define BEC and progenitor populations, as well as markers associated with both[8,16,23]. Mature BECs uniquely express the genes *CLDN10*, *CLDN4*, *CXCL2*, *LGALS2*, *MMP7*, *MUC1*, *MUC5B*, *TROP-2*, *TFF1*, *TFF2*, *TFF3* and *TSPAN8* as determined by our study and MacParland et al.[23] (Fig. 7). *CLDN4*, *MMP7*, *MUC1*, *TROP-2* and *TSPAN8* interestingly are also enriched in mouse BilPDs over HepPDs suggesting a gene signature of biliary specific lineage[8] (Fig. 7). We further identified a signature defining human hybrid progenitor cells, distinct from mature BECs, immature hepatocytes and mature hepatocytes. Genes, including *CAV1*, *CLDN6*, *GPRC5B*, *MCAM*, *NCAM* and *STAT4* are restricted to expression in HHyPs (Fig. 7). Several of these genes, including *MCAM* and *CAV1* are enriched in bipotent HepPDs that arise in mice after chronic liver injury[8]. Our study, therefore, has captured a transcriptional signature of previously undefined human foetal HHyPs, comparable to hepatic progenitor-like cells that arise in mice during liver injury.

## Discussion

In this study, we captured the molecular identity of distinct parenchymal and supporting cell populations in both foetal and adult human liver using scRNA-seq. We identified the transcriptional signature of a foetal human liver HHyP population, validated its presence in primary human liver samples, and showed its bi-lineage differentiation potential in vivo. In situ, HHyP cells could be anatomically differentiated from cholangiocytes/BECs that populate intra-hepatic BDs by using markers identified from our transcriptional profiling including TROP-2. Our work defines precise transcriptional changes during hepatic and biliary lineage commitment and provides evidence to suggest that hybrid hepatobiliary progenitor cells previously identified in mice also exist in humans during foetal development.

Previous work has proposed human liver stem/progenitors are bi-potent cells capable of repopulating both hepatocytes and cholangiocytes/BECs during development and injury[21,39]. These cells were reported to be EpCAM+/NCAM+, reside in the DP of human foetal liver and be retained in the Canals of Herring of normal adult livers[10,40]. Transcriptionally, they co-express genes associated with both hepatic and biliary lineage. Recent studies have captured similar hybrid-like liver progenitor cells in mouse and human adult liver tissue. In mouse, bi-potent progenitors have been traced from chronically injured hepatocytes[8] and mature BECs[3,4]. In humans, hepatocyte trans-differentiation post injury gives rise to hybrid cells[41]. Finally, in vitro small-molecule reprogramming of both mouse and human mature adult primary hepatocytes can generate a population of proliferating bipotent liver progenitor cells[14–16]. Despite validating their in vivo efficacy, little effort has been put into understanding the developmental origin of human liver progenitor cells and therefore how physiologically relevant they may be in a clinical setting. Using scRNA-seq, we identified cells with hybrid hepatobiliary characteristics within the human foetal liver EpCAM+/NCAM+ FACS population. These cells transcriptionally resemble the periportal hybrid progenitors previously identified in mice, being positive for *SOX9*, *HNF1B* and *KRT19*, but also *ALB*, *APOE* and

*TF*[6,8,9]. *HNFA* expression was also confirmed by in situ mRNA staining. They also express BEC/progenitor markers *CD24*, *CD133*, *CLDN3*, *FGFR2*, *KRT7* and *SPP1*. How these hybrid cells arise is contentious, having been previously attributed to hepatocyte plasticity[2,8], cholangiocyte trans-differentiation[4,9] or representative of a quiescent undifferentiated/reserve state in uninjured postnatal liver[6,10].

We found that HHyPs were distinct from foetal hepatoblasts which are AFP+/DLK+. Interestingly, recent scRNA-seq data on mouse liver was used to propose that hepatocyte and cholangiocyte lineages originate from AFP+/DLK+ hepatoblasts, and not from HHyPs[42]. However, the authors positively selected for DLK expressing cells, and therefore excluded EpCAM+/DLK−cells we identified. This cell capture strategy may explain the discrepancy with our results. Our cell-sorting protocol on the other hand is likely enriching for progenitors. Accordingly, we captured only a very limited number of mature ALB−/SOX9+/KRT7+ BECs. Intriguingly, our sorting strategy isolated a population of EpCAM+ cells highly similar to foetal HHyPs, expressing hepatic markers *ALB*, *HNF4A*, *RBP4* and biliary markers *SOX9*, *KRT19* and *KRT7*. However, these cells were negative for mature BEC markers identified in the recent Mac-Parland et al.[23] scRNA-seq study of human liver, including Trefoil factors (TFF)1–3 and mucins *MUC1*, *MUC3A* and *MUC5B*. This study however, also identified ALB expression in a subpopulation of BECs[23]. This raises the possibility that adult HHyPs identified in our study may indeed represent a population of mature BECs distinct from ALB−/TFF1+/TFF2+/TFF3+ BECs. TFFs have been previously reported to be heterogeneously expressed across different intrahepatic BDs[43,44]. Furthermore, it has recently been reported that adult mice BDs contain distinct BEC populations with differing proliferative capabilities[45]. Our findings suggest that this could be interpreted as revealing BEC heterogeneity in healthy liver. It is possible that these populations and the mechanisms that govern their behaviour regulate the inherent plasticity of parenchymal cell populations towards mature hepatic and biliary lineage in response to injury[4,8,14,15,41,46]. A key difference between foetal and adult HHyPs is the expression of TROP-2, also expressed in mature BECs. TROP-2 is a closely related family member of EpCAM thought to be absent or weakly expressed in normal liver tissue and enriched only at times of injury[30] or cancer[31]. Our data clearly show this is not the case in humans, with TROP-2 being expressed in adult BDs during normal homoeostasis. Our data further suggest that TROP-2 expression marks HHyP commitment towards biliary lineage, consistent with recent mouse data[8]. While this supports the notion that we have captured distinct populations of adult BECs, our analysis also confirms that adult HHyPs are transcriptionally close to foetal HHyPs validated in vivo for lineage potential and mouse 'oval cells'. Therefore, we cannot say conclusively that these cells are a specific mature BEC population without further functional validation, a collective problem the field of human liver progenitor studies faces. It is likely, however that, these populations respond differently to different injury stimuli. Understanding the phenotypes of distinct biliary populations is important when considering that the management of hepatic diseases is often hindered by difficulties in identifying their cellular origin. Given the frequent overlap in markers between multiple parenchymal cell types, it is crucial to accurately determine new histological markers.

Using the renal capsule of immunodeficient mice, we demonstrate the proliferative and hepatobiliary hybrid lineage potential of freshy isolated foetal human HHyPs. Investigating true functional stemness of freshly isolated progenitor cells for clonality, bi-potency and rescue was unfeasible in this study due to the low-cell numbers obtained post FACS sorting from rare human tissue.

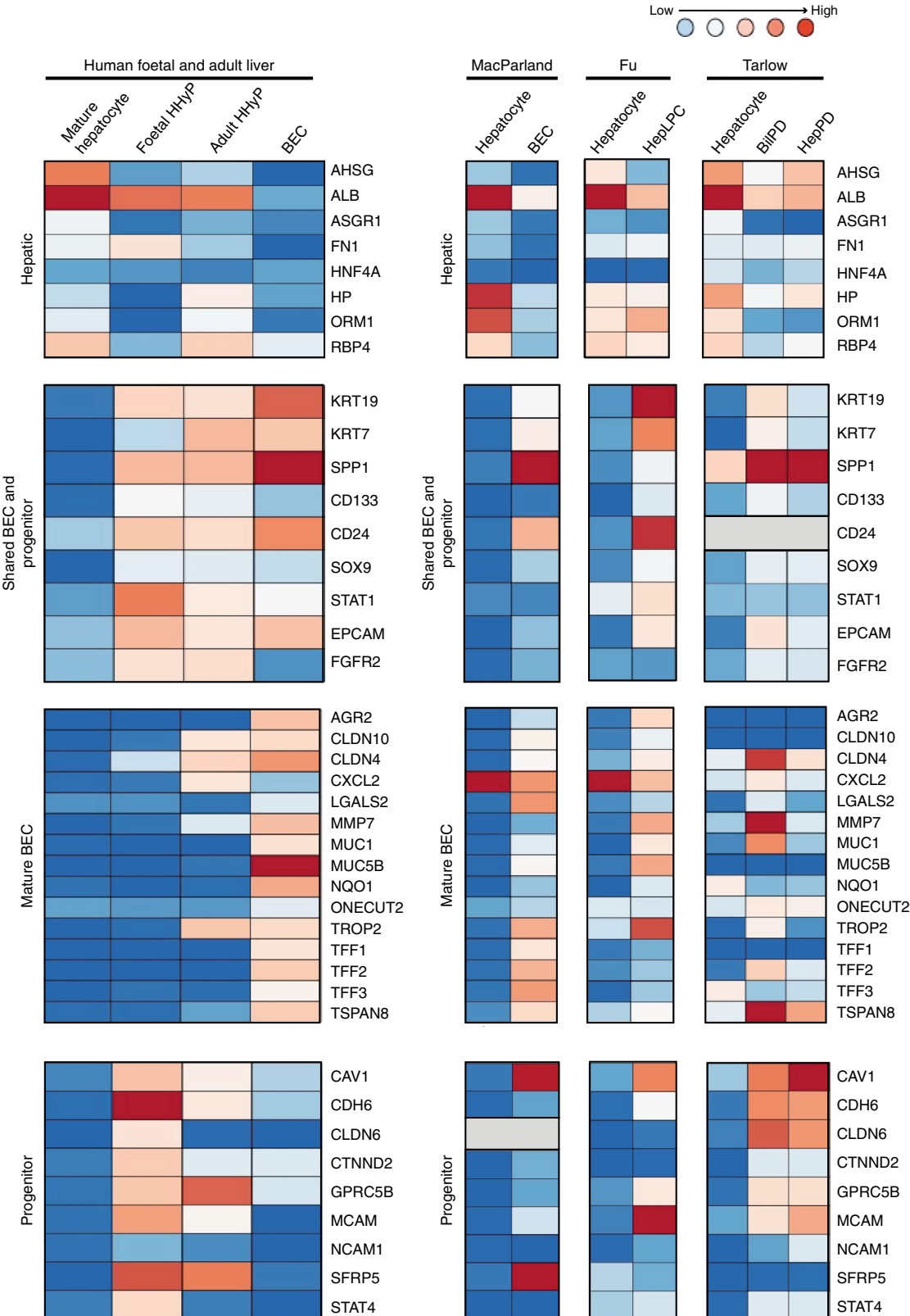

**Fig. 7** Collated marker expression in hepatic progenitor populations. Heatmaps of selected marker expression in populations identified in this study with different cohorts of liver progenitor-like cells, biliary epithelial cells (BECs) and primary hepatocytes. Shown is gene expression patterns of publicly available sequencing data from GEO (https://www.ncbi.nlm.nih.gov/geo/) or literature mined. Bulk RNA-seq data sets, include reprogrammed human hepatic liver progenitor cells (hepLPCs-Heps, GSE105019) and primary hepatocytes (pHs, GSE105019) from Fu et al.[16] and mouse hepatocyte-derived proliferative ducts (HepPD, Tarlow et al. 2015) and biliary-derived proliferative ducts ((BilPD) (GSE55552), Tarlow et al. 2015)[8]. ScRNA-seq data sets, include human BECs (GSE115469) from MacParland et al.[23]. Expression graded as high (red) to low (blue) relative to individual data sets

Recent studies focused on the translational potential of progenitor cells for liver repopulation required ex vivo expansion to produce sufficiently high numbers for transplantation into multiple animals, injured with enough severity such that an environment conducive to engraftment of donor cells is induced[10,21,40,47]. To avoid physiological variables that compound this approach, we elected to implant cells directly after FACS to experimentally match cells we assayed in vivo with those defined through scRNA-seq. This approach also completely removes results obtained as a consequence of ex vivo culture. We chose, therefore, to implant the foetal HHyP cell population into a non-hepatic space previously reported to be permissive for evaluation of the developmental potential of small numbers of human cells[32,33,37,48,49]. While this assay limits the functional validation of HHyP clonality and functional bi-potency of liver injury studies, this allowed us to investigate proliferative and hepatobiliary lineage potential of well-defined and freshly isolated rare human liver progenitor populations without in vitro expansion prior to transplantation[10,21,40,47,50]. Using this approach we show that foetal HHyPs within the EpCAM$^+$/NCAM$^+$ FACS population previously shown to harbour human foetal progenitor cells[10], are capable of expanding into FAH and HNF4A expressing hepatocytes within the kidney capsule environment. Specifically, EpCAM$^+$/NCAM$^+$/MCAM$^+$ FACS isolated human foetal liver HHyPs were capable of expansion within the renal capsule, and shown to be ALB$^+$/CK19$^+$ confirming the hepatobiliary hybrid phenotype of this population. Cells isolated from EpCAM$^+$/NCAM$^-$ foetal human livers did not engraft efficiently beneath the renal capsule. Our scRNA-seq analysis predicts this population is likely enriched for foetal immature hepatocytes, and may reflect their inability to expand in this niche. In our scRNA-seq analysis MCAM is identified as specifically enriching the foetal and adult HHyP phenotype, absent from expression in mature TTF1$^+$/TTF2$^+$/TTF3$^+$ BEC populations and from foetal hepatocytes. Furthermore, in mouse chronic liver injury studies, MCAM is enriched in hepatocyte derived proliferative ducts, over those derived from mature BECs alongside AHSG, ALB, CAV1 and RBP4[8]. Interestingly, MCAM (CD146) has previously been reported to be expressed on hepatic stellate cells that closely interact with human hepatic stem cells, as well as many different cell types including aggressive epithelial cancers[10,51,52]. MCAM forms homotypic cell–cell interactions, and therefore may play a key role in the niche–progenitor interface and regulating the interactions of foetal HHyPs with cells that comprise the stem cell niche during human liver development. Our in vivo findings validate our initial transcriptional labelling of FACS isolated putative foetal liver progenitors as hepatobiliary in nature. It will be of great interest for future research to investigate the presence of an expanded HHyP phenotype in human hepatobiliary disease tissue for fresh isolation in greater cell numbers to more specifically address questions on clonality, bi-potency and functional rescue potential.

Previous attempts to address fundamental questions surrounding the presence and nature of liver stem/progenitors have been hampered by the lack of correlation between mouse and human models and the limited resolution of lineage tracing strategies. By contrast, the combination of carefully selected human tissue and single-cell analysis performed here has facilitated the identification of a hybrid progenitor population from human foetal liver. Furthermore, these cells are transcriptionally distinct from mature human BECs. In our study we present an accurate gene expression profile of human foetal liver progenitor cells that the field can benchmark against for a biologically relevant phenotype. In-depth characterisation of the mechanisms regulating the behaviour of a hepatobiliary hybrid progenitor population will help advance our understanding of human liver development and disease.

## Methods

**ScRNA-seq cell sorting and cDNA library preparation**. All human tissues were collected with informed consent following ethical and institutional guidelines (Stanford, US and Kings College London, UK). Freshly isolated adult hepatocytes were obtained from Triangle Research Labs, while foetal livers were obtained from the Human Developmental Biology Resource of University College London. Human foetal tissue was dissociated by Collagenase XI enzymatic dissociation for 25 min at 37 °C with agitation. Samples were stained with the following primary antibodies, CD235a (349104, FITC, mouse; Biolegend), CD45 (304050, BV711, mouse; Biolegend), EpCAM (324208, APC, mouse; Biolegend), NCAM (362524, PE, mouse; Biolegend), ASGPR1 (563655, PE, mouse, BD Pharmingen™), TFRC (334106, PE mouse; Biolegend), MCAM (361005, PE, mouse; Biolegend), TACSTD2 (363803, PE, mouse; Biolegend) all at 1:100 dilution and incubated for 30 min at 4 °C. DAPI (D1306, ThermoFisher Scientific) at 1:1000 dilution was used for live/dead staining. Single-cell sequencing was performed using SmartSeq2[53]. Briefly, cells were sorted using a BD FACS Aria II instrument and deposited as single cells in 96-well plates, pre-loaded with lysis buffer (1% Triton X-100, 1 mM dNTP, 1 μM oligo-dT30, 1:1.2 × 106 ERCC ExFold RNA spike-in, Recombinant RNase Inhibitor (2313B, Takara Clontech). RNA was converted into cDNA using SMARTScribe Reverse Transcriptase (639538, Takara Clontech) and amplified for 21 cycles (Kapa HiFi HotStart ReadyMix 2×, KK2602, KAPA Biosystems). Successful single cell libraries were identified by capillary gel electrophoresis (DNF-474-1000, High Sensitivity NGS Fragment Analysis Kit, AATI) and converted into sequencing libraries using a Nextera XT DNA Sample Preparation Kit (FC-131-1096, Illumina). Barcoded libraries were pooled and subjected to 75 base pair paired-end sequencing on a Illumina NextSeq 2500 instrument.

**DNA sequencing and analysis of single-cell transcriptomes**. Raw sequencing reads were aligned using STAR and per gene counts were calculated using HTSEQ[54,55]. Gene counts were further analysed using the R package SCATER for pre-processing, quality control and normalisation[22]. To filter out unsuitable cells for scRNA-seq analysis we used median absolute deviations (MADs) on library size, total genes detected, mitochondrial gene percentage and artificial ERCC spike in percentage. Two deviations from the median were used to remove cell outliers. For library size and total genes the lower tail outliers were excluded. For mitochondrial and ERCC spike in percentage upper tail outliers were excluded. t-SNE, spearman's ranks co-efficient, hierarchical clustering and Student's $t$ tests were performed using custom scripts in R. t-SNE was performed on log10(TPM) normalised gene counts. For t-SNE analysis, the top 1000 variable features were used between all cells in the analysis. User-defined K-means clustering was performed in R. Optimal number of clusters was determined by peak cluster stability with silhouette analysis in R. For differential gene expression 1.50-fold change and $p$-value < 0.05 was used as a cut off determined by student t-test. Gene set enrichment analysis (GSEA) was performed on normalised scRNA-seq gene expression data through GSEA software run using the 'GO TERM' collection[56]. Pseudo lineage and lineage trajectory analysis was performed in R using the MONOCLE package[29].

**In situ RNA hybridisation**. Foetal and adult human liver samples were fixed in 10% formalin buffer saline (HT501128, Sigma Aldrich) for 2 days then dehydrated and paraffin wax infiltrated using Excelsior™ AS Tissue Processor. After embedding, sections (5 μm) were processed for in situ RNA hybridisation RNA-ISH using the RNAscope 2.5 High Definition (Red, 322350 ACD Bio) according to the manufacturers instructions. For single-plex staining the following probes were used: CDH6 (Hs-CDH6 403011), TACSTD2 (Hs-TACSTD2 405471), DCDC2 (HS-DCDC2 452911), ANXA13 (HS-ANXA13 542811), HNF4A (HS-HNF4A 442921), CD24 (HS-CD24 313021), CTNND2 (HS-CTNND2, custom), FGFR2 (Hs-FGFR2, 311171), (All ACD Bio). For Duplex RNA-ISH for transcript expression was performed using RNAscope® 2.5 HD Duplex Assay (322435, ACD Bio) according to manufacturer's instructions using the following custom probes: STAT1 (Hs-STAT1 469861, C2 channel change). Slides were counterstained with H&E QS (Vector Laboratories. Inc.). Mounted slides were imaged using NanoZoomer (Hamamatsu).

**Tissue processing and analysis for IF and IHC**. For IF staining, foetal liver samples were OCT (23-730-571, ThermoFisher Scientific) embedded, sectioned (5 μm) and fixed for 10 min in 4% w/v paraformaldehyde (PFA). Primary antibodies were used at the indicated dilutions: CDH6 (AF2715, sheep, 1:50; RnD systems), CLAUDIN-3 (83609 S, rabbit, 1:50, Cell Signalling), SOX9 (AF3075, goat, 1:100, RnD systems) and STAT1 (610115, mouse, 1:100; Transduction laboratories) at 4 ° C overnight. Cells were incubated with Alexa 647, Alexa 568, Alexa 488 conjugated secondary antibodies (all Life Technologies) and counterstained with DAPI (D1306, ThermoFisher Scientific). Images were acquired with a Nikon A1 confocal microscope (Nikon Instruments Inc.). Digital images were processed using NIS elements Advanced Research (Nikon) or ImageJ (https://imagej.nih.gov/ij/).

**IHC—tissue sections**. Paraffin-embedded foetal liver tissue was prepared as described for RNA-ISH. After embedding, sections (5 μm) were stained using mouse and rabbit specific HRP/AEC (ABC) Detection IHC Kit (abcam) using antibodies against EpCAM (ab7504, mouse, 1:100; abcam), CK19 (ab52625, rabbit,

1:200; abcam), CDH6 (ABIN950438, mouse, 1:100; Antibodies online) and STAT1 (610115, mouse, 1:100; Transduction laboratories) then counterstained with Eosin Y dye (ab146325, abcam). Mounted slides were imaged using a NanoZoomer (Hamamatsu).

**EpCAM⁺ foetal liver cell isolation.** Dissociated and filtered foetal liver cells were incubated with anti-CD326 (EpCAM) MicroBeads (130-061-101, Miltenyi Biotech) in 0.5% bovine serum albumin (BSA) with 5 U/ml DNASE1 (M0303, N.E.B.) and passed through a large cell separation column 2 × (130-042-202, miltenyi biotech). Total, EpCAM enriched and EpCAM depleted populations were FACS analysed for CD235a and EpCAM on a BD Fortessa cell analyser. For foetal liver intra-hepatic organoid formation (f-IHO) EpCAM enriched cells were pelleted at 300 × g for 5 min. The cell pellet was resuspended in a Matrigel® (354230, Corning) dome containing ~50,000 cells/dome in one well of a 48-well plate covered in 250 μl of liver expansion (LE) media. Media was changed every 48 h.

**Organoid culture conditions.** LE media was based on DMEM/F-12, GlutaMAX™ (10565018, Gibco, Life Technologies) supplemented with 1% N2 supplement (17502048, Gibco, Life Technologies), 2% B27 supplement (17504044, Gibco, Life Technologies), 20 mM HEPES (15630080, Gibco, Life Technologies), 1.25 mM N-Acetylcysteine (A7250, Sigma), 1% penicillin–streptomycin (15140122, Sigma Aldrich), 1% insulin:transferrin:selenium (ITS) (41400045, Gibco, Life Technologies) and the growth factors: 50 ng/ml EGF (AF-100-15, Peprotech), 500 ng/ml RSPO1 (120-38, Peprotech), 100 ng/ml FGF10 (100-26, Peprotech), 25 ng/ml HGF (00-39, Peprotech), 10 mM Nicotinamide (72340, Sigma), 25 μM RepSox (3742, Tocris), 10 μM Forskolin (1099, Tocris), 0.1 μM dexamethasone (1126, R&D Systems), 25 ng/ml Noggin (120-10 C, Peprotech), 1:100 Wnt3a (homemade, described in ref. [57], and 10 μM Y27632. HD media was based on DMEM/F-12, GlutaMAX™ supplement supplemented with 20 mM HEPES, 1.25 mM N-Acetylcysteine (A7250, Sigma), 1% penicillin–streptomycin (15140122, Sigma Aldrich), 1% ITS, 0.1 μM dexamethasone, 10 μM Y27632, 25 ng/ml HGF (00-39, Peprotech), 10 μM DAPT (D5942, Sigma), 2 ng/ml Oncostatin-M (295⁻OM-050, Bio-Techne) and 10 ng/ml BMP4 (314-BP-010, Bio-Techne). BD media was based on DMEM/F-12, GlutaMAX™ supplement supplemented with 20 mM HEPES, 1.25 mM N-Acetylcysteine (A7250, Sigma), 1% penicillin–streptomycin (15140122, Sigma Aldrich), 1% ITS, 0.1 μM dexamethasone, 50 ng/ml EGF, 500 ng/ml RSPO1 and 100 ng/ml DKK-1 (5439-DK, R&D systems), 3 μM Retinoic acid R2625, Sigma) and 10 ng/ml Activin-A (ActA, Q-kine). The culture medium was changed every 48 h.

**Passaging and staining organoids.** To split f-IHOs, Matrigel® was digested with Trypsin-EDTA for 15 min at 37 °C. Cell suspension was centrifuged at 300g for 4 min, washed once with William's E medium and resuspended in Matrigel® domes as described above. Organoids were typically passaged at a 1:4 ratio. For IF staining of organoids, Matrigel® was dissolved in cell recovery solution (11543560, ThermoFischer scientific) for 30 min at 4 °C and fixed for 30 min in 4% w/v PFA. Blocking was performed for 1 h in 3% donkey serum, 0.3% Triton and 0.1% DMSO. Primary antibodies were used 4 °C overnight at the indicated dilutions: CDH6 (AF2715, sheep, 1:50; RnD systems), EPCAM (ab7504, mouse, 1:100; abcam), KRT7 (ab9021, mouse, 1:100; abcam), HNF4A (ab92378, rabbit, 1:100; abcam), CK19 (ab52625, rabbit, 1:200; abcam), albumin (A80-129A, goat, 1:100; Bethyl), TROP2 (10428-MM02, mouse, 1:100; Stratech). DAPI was used at 1:1000 dilution as a counterstain. Alexa Fluor-conjugated secondary antibodies (all 1:500, Life Technologies). Images were acquired with a Nikon A1 inverted confocal microscope, a Nikon Ti spinning disk confocal microscope (Nikon Instruments Inc.) and a Leica TCS SP8 microscope (Leica Biosystems).

**In vivo transplantation of cells.** All experiments were performed in accordance with UK laws (Animal [Scientific Procedures] Act 1986) with approval of local ethics committee (King's College Animal Welfare and Ethical Review Board, London, UK) under a Home Office approved project licence. HHyPs or other foetal human liver cells were purified using FACS, resuspended in 20 μl of Matrigel and then injected using a Hamilton syringe underneath the renal capsule of 10–11-week-old immunodeficient NSG mice. Injected cells developed into a graft. The grafts were surgically removed for analysis after 4 weeks.

**IF staining of human xenografts.** After surgical removal human cell renal capsule grafts were fixed in 10% formalin buffer saline for 12 h with rotation, washed 3× in phosphate-buffered saline (PBS) then dehydrated and paraffin wax infiltrated using Excelsior™ AS Tissue Processor. For IF staining, antigen retrieval was performed for 20 min in 1× citrate buffer pH 6.0 (C9999, Sigma Aldrich). Specimens were permeabilised with 0.1% Triton in 1% bovine serum albumin (BSA) solution for 1 h, and blocked in 10% Donkey serum, 1% BSA in PBS for 2 h. Primary antibodies were used 4 °C overnight at the indicated dilutions: FAH (20-0042, rabbit, 1:100; Yecuris) CK19 (602-670, rabbit, 1:100; Abbomax), albumin (A80-129A, goat, 1:100; Bethyl), TROP2 (10428-MM02, mouse, 1:100; Stratech), HNF4A (ab92378, rabbit, 1:100; abcam). DAPI was used at 1:1000 dilution as a counterstain. Alexa Fluor-conjugated secondary antibodies (all 1:500, Life Technologies). Images were acquired with a Lecia DM6B upright microscope (Leica Biosystems).

## Data availability
The authors declare that all data supporting the findings of this study are available within the article and its Supplementary information files or from the corresponding author upon reasonable request. Raw RNA-seq data have been deposited in the Gene Expression Omnibus (GEO) database under accession code GSE130473.

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

## Acknowledgements

The research was funded/supported by the NIHR Clinical Research Facility at Guy's & St. Thomas' NHS Foundation Trust, UK; NIHR Biomedical Research Centre based at Guy's and St. Thomas' NHS Foundation Trust, UK; King's College London, UK; Medical Research Council (MRC) (MR/L006537/1), UK; National Institutes of Health (R01AI099245 and U19AI109662), US. We are grateful to the BRC Flow Cytometry Facility, KCH NHS Foundation Trust for advice and technical assistance. The views expressed are those of the author(s) and not necessarily those of the NHS, the NIHR or the Department of Health. The project has received funding from the European Union's Horizon 2020 Research and Innovation Programme under the Marie Sklodowska-Curie Grant agreement number 705607. We would also like to thank Prof. Fiona Watt, Simon Broad and Eamonn Morrison for their support. Several figures were created using Servier Medical Art templates, which are licensed under a Creative Commons Attribution 3.0 Unported License; https://smart.servier.com.

## Author contributions

S.T.R., J.S., H.N. and S.Q designed the study. J.S., D.K., S.S.N., M.P.S., B.O., S.B., A.K. and R.S conducted the experiments. R.Y., A.B., S.D. and D.J.W harvested and processed tissue for scRNA-seq analysis. J.S, D.J.W., G.K., G.E., A.V. and S.T performed the computational analysis. J.S., D.K., S.S.N., M.P.S., R.M. and T.V performed the histology. J.S., D.K., D.J.W., S.S.N., M.P.S., B.O. and S.T.R analysed and interpreted the data. J.S, D.K. and S.T. R prepared the paper with critical revision from all authors.

## Additional information

**Competing interests:** The authors declare no competing interests.

