## [Peer Review File · Nature Communications]

Reviewers' Comments:

Reviewer #1:

Remarks to the Author:

Single cell analysis of human liver identifies an anatomically restricted hepatobiliary hybrid progenitor population

The Rashid team have performed single cell analysis of human liver from foetal and adult liver. They are particularly interested in the biliary/progenitor cell compartment. This has been a very controversial area even in mouse and so good human studies are required in this area.

This manuscript has a strong single cell / genomics aspect but the functional validation is lacking. This significantly weakens the paper.

The main finding is the genomic description of ALB+/AFP-/KRT19+/SOX9+ hepatobiliary hybrid progenitor (HHyP) cells that are found in embryonic and adult tissues and distinct to human adult hepatocytes, biliary cells and foetal hepatocytes.

I have no specific comments on this part of the work – it seems to be well controlled and from a technical aspect carefully analysed from what I can see.

But- the lack of functional validation is important.

They grow organoids from human foetal liver and expand these as biliary like cells or differentiated hepatocyte like cells. They see some marker changes here (TRP-2 immunostaining). This is not sufficient to demonstrate the claims in the paper. I.e. "These data provide new evidence to suggest that hybrid hepatobiliary progenitor cells previously identified in mice also exist in humans and likely play critical roles in liver development and repair". To make this claim they should show functional data- I realise these are human cells but as a minimum, I would look for in vivo repopulation data of the different cell populations in immunodeficient mice.

The single cells has been used to identify novel markers which can be used to sort the cell populations from adult and fetal liver- do the resulting populations have different regenerative and differentiation potential? This has not been shown.

For a frontline journal such as this, I would expect functional data even with human cell analysis.

The human immunostaining data is pretty weak- high and low power sections should be shown for a variety of diseases and with significant "N" numbers. Dual staining and confocal analysis would help.

The authors institute is a huge adult and paediatric liver centre and so samples should very easy to come by...

Reviewer #2:

Remarks to the Author:

In the present study Segal et al. performed single cell RNA sequencing analysis for EpCAM+ biliary epithelial cells (BECs)/hepatic progenitor cells and/or hepatocytes from human fetal and adult livers. The results clearly indicate the heterogeneity among the EpCAM+ cell populations and the presence of hepatobiliary hybrid progenitor populations that are positive for both Albumin (ALB) and biliary markers, and have also identified TROP2 as a marker that can discriminate committed BECs from more immature progenitor-like cells. The study is quite informative and should be of significant interest and importance, providing an excellent platform for future advancement in the field of liver biology and pathology. Several concerns should be addressed before its acceptance for publication in the journal.

Figure 1, C and E. Expression profiles for EpCAM and NCAM should also be indicated for reference (could be in the supplementary figure).

Figure 3, A and C. While the pseudo-lineage analysis here clearly demonstrated the presence of divaricated lineage relationships, it cannot be conclusive regarding the “direction” of the differentiation/maturation process (or precursor–product relationship) in each of those lineages in principle. Thus, it is also and equally possible that the seeming “progenitor” cells in pseudo-states 1 and 2 are actually derived from pseudo-states 3 and/or 4. Indeed, their finding that the transcriptional profile in hybrid progenitor cells were similar to that in HepPDs identified in mice (Tarlow et al., 2014), which are bi-potent progenitor cells originating from mature hepatocytes, strongly support this possibility. The authors should address this issue in their manuscript, and revise those figures to clarify that the lineage relationships (arrows) can be bi-directional.

Figure 4. Expression of ALB, KRT19, SOX9, and NCAM in HHyP should also be examined and confirmed anatomically.

Figure 5, B–D. Those images should be accompanied with co-staining (or staining in adjacent sections) for conventional markers for the bile duct and the ductal plate, such as EpCAM, KRT7, and KRT19. It should also be clarified whether the gene expression in C ii and D, right panels, was in the “limiting plate” (i.e., hepatocytes) or rather in the terminal ductules.

Ex vivo culture experiments (Figure 6) using bulk EpCAM+ cells do not make any sense in order to support the authors’ claim regarding the heterogeneity and the presence of bi-potent progenitor subpopulations. EpCAM+ cells should be prospectively subdivided based on the expression of suggested markers, such as NCAM, CDH6, and TROP2, and then subjected to the culture to validate the presence/absence of the differentiation potentials toward hepatocytes and/or cholangiocytes in each subfractions. The fact that TROP2+ cell were present upon initial seeding but lost after passaging (Fig. 6A) is not conclusive as to whether TROP2 expression was indeed associated with an exit from a progenitor-like state, or the TROP2+ cells merely lost their expression of TROP2 upon culture under that particular condition.

Figure 7. Those images should be accompanied with co-staining (or staining in adjacent sections) for conventional markers for the ductular reaction, such as EpCAM, KRT7, and KRT19.

In relation to Figure 7B, it should be of significant interest to examine and compare the TROP2 expression in the ductular reaction of human liver with cholestatic disease conditions, where the biliary phenotype in the bile ducts and ductules is supposed to be maintained.

Abstract, line 10. The present study is still not conclusive as to whether the hybrid progenitor cells are indeed bi-potent. Thus, the term “bi-potent progenitors” written here is misleading and should be changed to “hybrid progenitors” or “bi-phenotypic progenitors”, etc.

Reviewer #3:

Remarks to the Author:

General overview

With the goal of addressing a long-standing question regarding the potency of a specific type of liver stem cell, Segal et al. present a single-cell RNA-seq (scRNA-seq) analysis performed on cells from fetal and adult human liver. Specifically, the authors aimed at studying the cellular heterogeneity during human liver development and propose the existence of a hepatobiliary hybrid progenitor (HHyP) population found both in fetal and adult liver tissues. The authors claim that these cells represent a bipotent biliary and hepatic progenitor population in human, in line with findings previously presented for mouse. They identified TROP-2 expression to be distinctive for biliary lineage commitment of HHyPs, proposing it therefore as a marker gene for stratifying HHyPs

from bile duct cells that both reside in spatial proximity within liver periportal regions. In addition, the authors demonstrate that TROP-2 expression is distinctive for cells involved in proliferation of duct structures during liver injury and therefore might be used in clinical diagnostics.

Overall, the paper is well written and at first glance appears to communicate a clear message. However, a second, more in-depth reading of the manuscript triggered concerns regarding the robustness of the stated results. Most importantly, the authors do not provide the necessary data to formally prove that HHyPs can give rise to mature hepatocytes in-vivo. Therefore, it is at this point unclear whether these cells can be strictly called "bi-potent". Below, a list of major concerns / comments is provided that the authors will hopefully consider to improve the overall clarity of their findings.

Major comments:

1. The t-SNE map surprisingly shows clearly distinct and defined populations without any cells representing a transition from one state to another, such as the commitment from the progenitor cells to the hepatocytes. It is also unclear why cholangiocytes are not present in the data set. One potential explanation is that a too low number of cells was analyzed to faithfully represent the stages of differentiation and various populations. Technical issues can also not be excluded. It is in this regard puzzling why the authors did not attempt to integrate already available scRNA-seq datasets for human fetal and adult liver (e.g. Camp et al. *Nature*, 2017) to not only evaluate the validity of their data but also to increase the size and thus resolution of their dataset. Such an integration is highly recommended.
2. The coherence between the FACS sorting strategy and the assignment to the different clusters is questionable. Indeed, many hematopoietic cells were detected in the data set while they are supposed to be EpCAM⁻. Furthermore, it needs to be explained why a considerable number of fetal and mature hepatocytes, that are normally EpCAM⁻, were identified within the EpCAM⁺ population scRNA-seq data.
3. The population of HHyPs is assumed to be similar in adult and fetal tissues, however, their transcriptomic profiles seem to indicate that they represent two distinct populations. A more in-depth analysis of their shared (or distinct) characteristics is therefore crucial.
4. The reliability of the results supporting the existence of a hepatobiliary hybrid progenitor population, as identified in mouse, is not clear. For example, figure 1E displays the distribution of expression of a subset of genes in HHyP versus the few cholangiocytes from the scRNA-seq data, compared to bulk results from HepPD and BILPD in mouse. The authors concluded that the HHyPs have very similar expression patterns to HepPDs. First of all, the analysis would benefit from a more global comparison and not only based on a few genes. The expression of the orthologous top marker genes of the HepPDs should be shown in the human data. Furthermore, it is not possible to directly compare log(TPM) values from different data sets and even less from bulk versus single-cell RNA-seq. It would be more informative to indicate for example the fold changes and if the differences are significant. The HHyPs can also be compared against different cell types and not only cholangiocytes. The authors could also investigate the possibility to apply powerful batch correction techniques such as scmap (Kiselev et al., *BioRxiv*, 2017) to project the single-cell results onto the referenced cell types from the bulk RNA-seq.
5. While mature cholangiocytes are ALB⁻, this marker can still be expressed in biliary progenitor cells. One is therefore left to wonder what exactly the data is that excludes that HHyPs are not mere biliary progenitors? In line, AFP, which is expressed in hepatoblasts and embryonic hepatic cells, is not expressed in any fetal HHyPs. This points to a similarity between HHyPs and mouse "oval cells", being also AFP⁻. The latter cells were long believed to be bi-potent progenitors but clear evidence for their capacity to produce mature hepatocytes is still lacking (reviewed in Kopp et al. *Nature Cell Biology*, 2016). In addition, TROP-2 was previously found to be distinctive for mouse "oval cells", which differentiated these cells from cholangiocytes (Okabe et al.,

Development, 2009 (doi: 10.1242/dev.031369)). Taken together, it would be valuable with respect to understanding the true nature of HHyPs if the authors could perform a comparison of transcription profiles between HHyPs and rodent oval cells.

6. The authors used 3D organoids to validate the functional relevance of EpCAM+ sorted cells. To prove the functional proximity of cultured cells used for in-vitro differentiation to HHyPs, it is necessary to provide a comparative analysis (transcriptomics). In addition, it would be valuable if the authors would support their IF data with gene expression profiling to highlight that TROP-2 negative cells are capable of giving rise to both cell types.

7. The evidence supporting the bipotency of HHyPs is, as already indicated, rather scarce. Indeed, the conclusions are mainly based on the pseudo-time trajectory analysis. However, the results of the latter analysis are not linked to the mature hepatocytes and cholangiocytes questioning the biological relevance of the branching event. How would this trajectory look like if mature cells would have been included? One can also wonder why no fetal cells are located along the two branches as duct cells and hepatocytes are also present in fetal livers, questioning once again the biological relevance of this analysis as well as the similarity between adult and fetal HHyPs. Thus, the putative bipotency in fetal liver remains questionable. An analysis with Monocle including mature cells and/or only on adult HHyPs could potentially clarify these questions. The results of StemID are also not described well enough to be informative for the reader.

8. The fetal and adult HHyPs seem to constitute distinct populations and the authors don't provide sufficient evidence supporting their similarity (see also point 3). Moreover, TROP-2 expression, identified as a key marker to differentiate cells committed toward the biliary lineage from uncommitted HHyPs, seems to also separate fetal (TROP-2-) and adult cells HHyPs (TROP-2+). It is therefore questionable whether the lineage tracing analysis of mixed HHyPs (fetal and adult) is unbiased. Specifically, TROP-2 was identified as a ductal marker in the pseudo state 3. Since this population consists of only adult cells, it might highlight a developmental stage rather than a specific cellular commitment state. The analysis might be more convincing if performed separately for both fetal and adult HHyPs and if involving more cells. It would thereby be equally valuable if independent analytical validation would be sought for the trajectory analysis using for example the RNA velocity tool (La Manno et al., Nature, 2018).

9. The single-cell RNA-seq analysis workflow description is rather shallow, rendering it difficult to evaluate its quality / robustness.

Other comments:

1. The potential cholangiocytes are not indicated on the t-SNE map.
2. The number of cells for each cluster is not stated.
3. The genes on the heat map are often not indicated.
4. The authors do not investigate at all pseudo state 5.

Reviewer #1

The Rashid team have performed single cell analysis of human liver from foetal and adult liver. They are particularly interested in the biliary/progenitor cell compartment. This has been a very controversial area even in mouse and so good human studies are required in this area. This manuscript has a strong single cell / genomics aspect but the functional validation is lacking. This significantly weakens the paper. The main finding is the genomic description of ALB+/AFP-/KRT19+/SOX9+ hepatobiliary hybrid progenitor (HHyP) cells that are found in embryonic and adult tissues and distinct to human adult hepatocytes, biliary cells and foetal hepatocytes.

I have no specific comments on this part of the work – it seems to be well controlled and from a technical aspect carefully analysed from what I can see. But- the lack of functional validation is important. They grow organoids from human foetal liver and expand these as biliary like cells or differentiated hepatocyte like cells. They see some marker changes here (TROP-2 immunostaining). This is not sufficient to demonstrate the claims in the paper. Ie “These data provide new evidence to suggest that hybrid hepatobiliary progenitor cells previously identified in mice also exist in humans and likely play critical roles in liver development and repair”. To make this claim they should show functional data- I realise these are human cells but as a minimum, I would look for in vivo repopulation data of the different cell populations in immunodeficient mice.

Reviewer 1 - Point 1: Clarification on the organoid experiment as validation of scRNA-seq markers defining biliary lineage commitment.

We thank the reviewer for pointing out that the organoid experiments did not provide sufficient evidence to make claims that HHyPs play critical roles in liver development and repair. We have addressed this in 3 ways:

- (i) Providing additional in vivo functional data (see reply to editor above and reply to reviewer below)
- (ii) Changing the text of our manuscript (see below) and
- (iii) Contextualizing the organoid experiments better (see below)

Organoid experiments were in fact performed to validate lineage commitment markers identified through single cell analysis. Hybrid EpCAM⁺/SOX9⁺/HNF4A⁺ organoids, become TROP2⁻/CK7⁻/ALB⁺ when pushed towards hepatic and TROP2⁺/CK7⁺/ALB⁻ when pushed towards biliary lineages (Fig. 4F). This confirmed *ex vivo*, our *in situ* findings that TROP-2 is a key marker to distinguish TROP-2⁻ human foetal liver hybrid progenitors from TROP-2⁺ committed BECs present in bile ducts. These results support our claim that “the expression signature of foetal HHyPs from our scRNA-seq dataset can be utilized to distinguish a human liver progenitor from biliary and hepatic committed cells, as clarified in

the revised manuscript (Page 9, line 15) and not of HHyPs “playing a critical role in liver development and repair”.

The single cells has been used to identify novel markers which can be used to sort the cell populations from adult and fetal liver - do the resulting populations have different regenerative and differentiation potential? This has not been shown. For a frontline journal such as this, I would expect functional data even with human cell analysis.

Reviewer 1 - Point 2: Validating The differentiation potential of hepatobiliary hybrid progenitor cells sorted from human foetal liver.

We very much appreciate the perspective of the reviewer and address it as described in our reply to the editor above. In brief - we FACS sorted HHyPs from human foetal liver using novel markers identified by our single cell analysis to conduct *in vivo* functional studies (Fig. 5A). We used a marker panel of CD235a/CD45/EpCAM⁺/NCAM⁺/MCAM⁺ to isolate HHyPs (Fig. 5B) and demonstrated that only these cells were capable of expansion within the renal capsule of immunodeficient mice compared to controls (the same number of cells were transplanted) (Fig.5C). After 4 weeks, HHyP implants contained cells of both hepatic (FAH⁺, HNF4A⁺, ALB⁺) (Fig. 5E-F, , Fig. S7D) and biliary (CK19⁺, TROP2⁺) lineages (Fig. 5E,G) as well as cells retaining a progenitor phenotype (CK19⁺/ALB⁺) (Fig. 5E). These results provide new *in vivo* evidence of the lineage potential of foetal human HHyPs.

Fig. 5A-B: Schematic of strategy to isolate HHyPs from human foetal liver and implant into the renal capsule of immunodeficient mice for *in vivo* expansion (A). Gene expression overlaid on t-SNE profile of human foetal liver cells isolated in scRNA-seq analysis. ALB⁺ foetal HHyPs isolated from the EpCAM/NCAM⁺ FACS population are exclusively MCAM⁺. We therefore used EpCAM⁺/NCAM⁺/MCAM⁺ to FACS isolate HHyPs for *in vivo* implantation.

Fig. 5C: Matched cell numbers of EPCAM⁺/NCAM⁻/MCAM⁻ (Cell control) and EpCAM⁺/NCAM⁺/MCAM⁺ (HHyP) were implanted beneath the renal capsule of NGS mice. Only HHyP cells were capable of expansion with the renal capsule. Also included is the matched host kidney with no cells implanted.

Fig. 5D: IF staining of FAH (red) and HNF4A (green) in implant region of HHyPs 4 weeks post transplantation. Slides counterstained in DAPI (cyan).

Fig. 5E: IF co-staining of ALB (red) and CK19 (green) within HHyP implant regions 4 weeks post transplantation and matched control mouse adult liver. Slides counterstained in DAPI (cyan). Within the HHyP explant regions we identify ALB⁺/CK19⁺ cells confirming the presence of hybrid hepatobiliary cells. We also identify ALB⁺/CK19⁻ cells (red arrows), suggesting maturation from a bi-potent phenotype to hepatic lineage. We also confirm the anti-ALB antibody is human specific, and therefore cells of the explant are of human origin.

“The human immunostaining data is pretty weak- high and low power sections should be shown for a variety of diseases and with significant “N” numbers. Dual staining and confocal analysis would help. “

Reviewer 1 - Point 3: Requirement of higher numbers of disease sample staining.

We acknowledge the criticism made and agree that significantly higher numbers of disease samples would be required to validate this part of the manuscript. Given the amount of time / resource required to achieve such validation and that this is not the main focus of our narrative we conclude such an undertaking to be outside the scope of this current work and better suited to an independent follow up paper. Human disease immunostaining data has therefore been removed from the revised manuscript.

Reviewer #2

In the present study Segal et al. performed single cell RNA sequencing analysis for EpCAM+ biliary epithelial cells (BECs)/hepatic progenitor cells and/or hepatocytes from human fetal and adult livers. The results clearly indicate the heterogeneity among the EpCAM+ cell populations and the presence of hepatobiliary hybrid progenitor populations that are positive for both Albumin (ALB) and biliary markers, and have also identified TROP2 as a marker that can discriminate committed BECs from more immature progenitor-like cells.

“The study is quite informative and should be of significant interest and importance, providing an excellent platform for future advancement in the field of liver biology and pathology. Several concerns should be addressed before its acceptance for publication in the journal.

Figure 1, C and E. Expression profiles for EpCAM and NCAM should also be indicated for reference (could be in the supplementary figure).”

Reviewer 2 - Point 1: Expression profiles for EpCAM and NCAM in Figure 1, C and E.

We thank the reviewer for this request and agree that it brings useful context to the data. We have accordingly overlaid expression profiles of *EpCAM* and *NCAM1* on scRNA-seq t-SNE analysis as below and inserted those figures into Supplementary Figure 1D.

Fig. S1D: Transcript expression of *EpCAM*, *NCAM1* and *HNF4A* overlaid on the 2D t-SNE space of human liver scRNA-seq analysis. Expression is Log10(TPM).

“Figure 3, A and C. While the pseudo-lineage analysis here clearly demonstrated the presence of divaricated lineage relationships, it cannot be conclusive regarding the “direction” of the differentiation/maturation process (or precursor–product relationship) in each of those lineages

in principle. Thus, it is also and equally possible that the seeming “progenitor” cells in pseudo-states 1 and 2 are actually derived from pseudo-states 3 and/or 4. Indeed, their finding that the transcriptional profile in hybrid progenitor cells were similar to that in HepPDs identified in mice (Tarlow et al., 2014), which are bi-potent progenitor cells originating from mature hepatocytes, strongly support this possibility. The authors should address this issue in their manuscript and revise those figures to clarify that the lineage relationships (arrows) can be bi-directional.”

Reviewer 2 - Point 2: Bi-directional potential of HHyP pseudo lineage analysis

We thank the reviewer for this helpful comment and have addressed the points raised through a combination of editing both the figures and text as below.

- (i) The pseudo lineage plot has been revised to clarify that lineage relationships can be bi-directional between progenitor, hepatic and biliary populations due to previously identified plasticity between liver parenchymal cell types (Fig. S6A).
- (ii) Acknowledging the comment that HepPDs identified in the Tarlow *et al* (2014) study originate from mature hepatocytes after injury, we would like to re-emphasize that our study used human foetal liver samples to capture a developmental hybrid progenitor population, relative to uninjured adult populations. Indeed, our study would therefore support the suggestion that the HepPD phenotype derived from hepatocyte de-differentiation post injury is transitioning back towards a foetal-like HHyP phenotype, consistent with what has been shown in other organs (Yiu et al Cell Stem Cell 2018 PMID 29249464). We have accordingly revised the manuscript to say that pseudo-lineage analysis revealed distinct clusters of progenitor, hepatic and biliary lineage populations, without assuming directionality (Page 10, Line 2).

Fig. S6A: PCA plot of Monocle pseudo-time analysis on all cells of foetal and adult hybrid hepatobiliary progenitor (HHyP) cluster, coloured by pseudo state.

“Figure 4. Expression of ALB, KRT19, SOX9, and NCAM in HHyP should also be examined and confirmed anatomically.

Figure 5, B–D. Those images should be accompanied with co-staining (or staining in adjacent sections) for conventional markers for the bile duct and the ductal plate, such as EpCAM, KRT7, and KRT19. It should also be clarified whether the gene expression in Cii and D, right panels, was in the “limiting plate” (i.e., hepatocytes) or rather in the terminal ductules.”

Reviewer 2 - Point 3: Co-staining of HHyPs with conventional markers

We thank the reviewer for this request and agree that it brings added validity to our findings. As we are sure would already have been noted, the manuscript contained IHC and IF staining for conventional ductal plate and bile duct markers KRT19 (IHC, Fig. 4C) and CLDN3 (IF, Fig. 4B), alongside newly identified HHyP markers CDH6 and STAT1, that clearly localised to the foetal ductal plate and bile duct anatomical regions. Furthermore, we showed by *in situ* staining that the hepatic marker *HNF4A* is co-expressed with HHyP marker *STAT1* in the ductal plate region of human foetal liver (Fig. S5C).

We have now however also included

- (i) New *in situ* staining of *ALB*, confirming the anatomical location of *ALB* as being absent from bile ducts (Fig. S5B)
- (ii) An expanded panel of HHyP markers to further confirm their restricted expression to the ductal plate, including *CD24* a well-established liver progenitor marker (Fig.S5A)^{26,27}.
- (iii) IF co-expression data of SOX9 with markers CLDN3 and CDH6 to confirm their anatomical location in the ductal plate (Fig. S5C).
- (iv) EpCAM expression in our IHC staining panel as a conventional marker for human bile ducts and the ductal plate.

We feel it is also important to clarify that staining of biliary markers such as EpCAM, CLDN3, SOX9 and KRT19 are expressed in both ductal plate and bile duct structures. One of the most important findings of our study is that it identifies - for the first time as far as we know - the marker *TROP-2* as being exclusively expressed in bile ducts, and absent from hybrid liver progenitors.

B

ALB

Fig. S5B: RNA-ISH for *ALB* (red) on human 2nd trimester (15-21 pcw) foetal liver ductal plate (DP) and bile ducts (BD) regions. Scale bars represent 50 μ m. Scale bar of zoomed in region represents 25 μ m. Our staining confirms that *ALB* is highly expressed in the hepatic parenchyma, but absent from intrahepatic bile ducts, confirming our scRNA-seq analysis that HHyPs that are *ALB*⁺ are not isolated from foetal liver bile ducts.

A

CD24

FGFR2

DCDC2

CTNND2

Fig. S5A: RNA-ISH for *CD24*, *FGFR2*, *DCDC2* and *CTNND2* on human 2nd trimester (15-21 pcw) foetal liver ductal plate (DP) regions. These markers are all enriched in the HHyP cell population identified in human foetal liver scRNA-seq analysis, and restricted to the ductal plate as confirmed by *in situ* staining.

Di

Dii

Fig. S5D: Immunofluorescence (IF) staining of CDH6 (green) and CLDN3 (white) with SOX9 (red) co-expression in 2nd trimester human foetal liver (i) and zoomed in images of DP and BD regions. Slides counterstained in DAPI (cyan). DP and BD structures outlined white arrows. SOX9 is a conventional biliary marker. We confirm its co-expression in the ductal plate and bile ducts with newly identified HHyP marker CDH6.

For figure 5 B-D, adult *in situ* staining of bile duct and limiting plate structure's is now Fig. S6D. The staining we present clearly defines bile duct and 'limiting plate structures and are labelled as such.

Fig. S6D: Duplex RNA-ISH for CDH6, DCDC2 and ANXA13 (red) with STAT1 (blue) in adult human liver bile duct (BD) and limiting plate (LP) structures. Detail of the red dots expanded in the squares. We have clearly labelled the bile duct and limiting plate regions.

“Ex vivo culture experiments (Figure 6) using bulk EpCAM⁺ cells do not make any sense in order to support the authors’ claim regarding the heterogeneity and the presence of bi-potent progenitor subpopulations. EpCAM⁺ cells should be prospectively subdivided based on the expression of suggested markers, such as NCAM, CDH6, and TROP2, and then subjected to the culture to validate the presence/absence of the differentiation potentials toward hepatocytes and/or cholangiocytes in each subfractions. The fact that TROP2⁺ cell were present upon initial seeding but lost after passaging (Fig. 6A) is not conclusive as to whether TROP2 expression was indeed associated with an exit from a progenitor-like state, or the TROP2⁺ cells merely lost their expression of TROP2 upon culture under that particular condition.”

Reviewer 2 - Point 4: Clarification of the organoid experiment

We thank the reviewer for these comments and would ask to kindly refer to our responses above addressing Reviewer 1 points 1 and 2. In brief - our new data subdivides populations of EpCAM⁺ cells on NCAM and TROP2 (Fig. S7A-B) with EpCAM⁺/NCAM⁺/TROP2⁻ HHyPs found to be (i) more capable of expansion following implantation into the renal capsule of immunodeficient mice compared to equal cell numbers of EpCAM⁺/NCAM⁻/TROP2⁻ cells and (ii) able to form both hepatic and biliary lineages. TROP2⁺ populations (CD235a⁺/CD45⁻/EpCAM⁺/NCAM⁻/TROP2⁺) were (predictably) difficult to isolate. The combined data from *in vitro* (Matrigel) and *in vivo* (kidney capsule) experiments supports our conclusions that HHyPs identified are not foetal biliary cells but rather progenitor cells with both hepatic and biliary lineage differentiation potential.

Fig. S7: 2D t-SNE visualisation of single cells isolated from foetal and adult human liver coloured by TROP-2 gene expression. Expression is Log₁₀(TPM) (i). Schematic of hepatic and BEC lineage restriction of HHyPs based on NCAM/TROP-2 expression (ii). **(B)** Gating scheme for the isolation of distinct foetal human liver populations based on expression of CD235a, CD45, EpCAM, NCAM and TROP2. **(C)** Hematoxylin and eosin (H&E) staining in tissue cross-sections of HHyP explant regions for 4 week post renal capsule transplantation. Scale bars represent 250 μ m. **(D)** Immunofluorescence (IF) staining of FAH (red) in implant region of HHyPs post 4 weeks transplantation. Slides counterstained in DAPI (cyan). Scale bars represent 75 μ m.

“Figure 7. Those images should be accompanied with co-staining (or staining in adjacent sections) for conventional markers for the ductular reaction, such as EpCAM, KRT7, and KRT19. In relation to Figure 7B, it should be of significant interest to examine and compare the TROP2 expression in the ductular reaction of human liver with cholestatic disease conditions, where the biliary phenotype in the bile ducts and ductules is supposed to be maintained.”

Reviewer 2 - Point 5: Disease sample staining

We agree with reviewer 2’s comments that it would be of significant interest to examine TROP-2 expression in the ductular reaction of human livers with cholestatic disease. However, as per our reply to reviewer 1’s point 3, we have removed human disease staining from this study as in our opinion these (likely extensive) investigations are better addressed in a separate, follow on study.

“Abstract, line 10. The present study is still not conclusive as to whether the hybrid progenitor cells are indeed bi-potent. Thus, the term “bi-potent progenitors” written here is misleading and should be changed to “hybrid progenitors” or “bi-phenotypic progenitors etc”

Reviewer 2 - Point 6: Nomenclature

We have revised the text accordingly to encompass this helpful suggestion.

Reviewer #3:

General overview

With the goal of addressing a long-standing question regarding the potency of a specific type of liver stem cell, Segal et al. present a single-cell RNA-seq (scRNA-seq) analysis performed on cells from fetal and adult human liver. Specifically, the authors aimed at studying the cellular heterogeneity during human liver development and propose the existence of a hepatobiliary hybrid progenitor (HHyP) population found both in fetal and adult liver tissues. The authors claim that these cells represent a bipotent biliary and hepatic progenitor population in human, in line with findings previously presented for mouse. They identified TROP-2 expression to be distinctive for biliary lineage commitment of HHyPs, proposing it therefore as a marker gene for stratifying HHyPs from bile duct cells that both reside in spatial proximity within liver periportal regions. In addition, the authors demonstrate that TROP-2 expression is distinctive for cells involved in proliferation of duct structures during liver injury and therefore might be used in clinical diagnostics.

“Overall, the paper is well written and at first glance appears to communicate a clear message. However, a second, more in-depth reading of the manuscript triggered concerns regarding the robustness of the stated results. Most importantly, the authors do not provide the necessary data to formally prove that HHyPs can give rise to mature hepatocytes in-vivo. Therefore, it is at this point unclear whether these cells can be strictly called “bi-potent”. Below, a list of major concerns / comments is provided that the authors will hopefully consider to improve the overall clarity of their findings.

We thank reviewer 3 for their suggestions and acknowledge that we have generated a complex data set with numerous avenues of potentially highly informative follow up investigation. By addressing the reviewer’s comments below however we believe we have significantly improved the robustness and clarity of our results as suggested. In particular, we have gone to great lengths, as requested by the reviewer, to demonstrate HHyPs can give rise to mature hepatocytes in-vivo (please see reply to editor, reviewers 1 and 2 and Fig. 5).

Major comments:

1. The t-SNE map surprisingly shows clearly distinct and defined populations without any cells representing a transition from one state to another, such as the commitment from the progenitor cells to the hepatocytes.”

Reviewer 3 – Point 1A: Explanation of distinct populations within scRNA-Seq t-SNE map.

We thank the reviewer for their comment and also agree that it was surprising to see such a distinction amongst cell populations by t-SNE analysis (Fig. 1B). That observation is due to

- (i) the FACS sorting strategy employed which was in itself deliberately designed to isolate specific sub populations of foetal liver.
- (ii) the global t-SNE analysis representing all cells including non-parenchymal cell types, which have significantly distinct gene expression patterns (Table S1-5).

Since the importance of our FACS strategy in isolating distinct cell types may not have been made clear because our initial t-SNE plot was coloured phenotypically by cell-type, we have revised the figures to overlay the FACS populations on initial t-SNE analysis. This we believe provides a clearer understanding of the FACS strategy and its effectiveness in isolating distinct cell types (Fig. 1B).

Fig. 1B: 2D t-SNE visualisation of single cells isolated from foetal and adult human liver coloured by FACS gating population, shaped by tissue source. This validates how are FACS has been designed to isolate specific hepatic populations, including potentially rare progenitor populations.

Reviewer 3 – Point 1B: Cells representing a transition from one state to another

We agree with the reviewer’s comment that our initial t-SNE analysis does not clearly reveal any cells representing a transition from one state to another. However, pseudo lineage analysis on HHyP populations does in fact identify cells that are transitioning between hepatic and biliary states (Fig. S6) as was kindly noted by reviewer 2 “analysis here clearly demonstrated the presence of divaricated lineage relationships”. We do however acknowledge that our labelling was not as clear as it could have been and have accordingly revised the manuscript to make highlight the potential bi-directionality of these populations (page 9, line 21 – page 10, line 3). In summation we have clarified that we did identify cells with transitional phenotypes between different cell types.

“1.It is also unclear why cholangiocytes are not present in the data set. One potential explanation is that a too low number of cells was analyzed to faithfully represent the stages of differentiation and various populations. Technical issues can also not be excluded. It is in this regard puzzling why the authors did not attempt to integrate already available scRNA-seq datasets for human fetal and adult liver (e.g. Camp et al. Nature, 2017) to not only evaluate the validity of their data but also to increase the size and thus resolution of their dataset. Such an integration is highly recommended.”

Reviewer 3 – Point 1C: Presence of Cholangiocytes and available scRNA-seq datasets

We thank the reviewer for this comment and would first politely request to draw their attention to the single-cell RNA-seq study of adult human liver recently published in this journal²⁴. While the focus of that paper was on immune cell populations, the authors also identified a population of biliary epithelial cells (“BECs” / cholangiocytes) that we were able to integrate into our own analyses to generate distinct transcriptional phenotypes separating adult/mature BECs from adult HHyPs. We believe combining these two data-sets adds value beyond that of the Camp et al. Nature²⁸, 2017 data since both studies leverage fresh (not frozen) human samples containing adult/mature BECs. As a result, we were able to greatly improve the resolution and understanding of the different cell types identified (Fig. S3). We have accordingly changed the manuscript as below:

- (i) clearly labelled biliary epithelial cells (“BECs” / cholangiocytes) on the 2D t-SNE plot (Fig. 1E)
- (ii) added gene expression analysis of mature BEC markers (identified in this study as exclusively BEC over HHyPs) (Fig. 6) and
- (iii) integrated our data with the hepatocyte and BEC clusters from MacParland et al (2018)(Fig. S3)²⁴.

Fig. 1E: 2D t-SNE visualisation of single cells isolated from foetal and adult human liver coloured by cell type. Phenotypic labelling based on transcriptional analysis. Mature biliary epithelial cells (BECs) are coloured yellow.

Fig. 1C: Transcript expression of mature BEC markers TFF1 and TFF2 overlaid on the 2D t-SNE space of human liver scRNA-seq analysis. Expression is Log10(TPM).

Fig. S3: Comparison of HHyP gene expression with mature BECs. We performed data integration of our scRNA-seq analysis with recently published adult human liver data from MacParland *et al.* (2018)²⁴, which included a cluster of mature BECs. Observing the expression of significantly upregulated genes in the mature human biliary epithelial cell (BEC) cluster from MacParland *et al.* 2018²⁴ in foetal hepatocytes, adult hepatocytes, BECs, foetal HHyPs and adult HHyPs identified in this study (A) reveals a unique signature of markers expressed exclusively in BECs, including *TFF2*, *MUC5B*, *TFF3*, *LGALS2*, *AGR1* and *NQO1*. These markers are enriched in mature BECs we identify in our study. More importantly they clearly distinguish a BEC phenotype from HHyPs, significantly increasing the resolution of our dataset and validating our finding of hybrid progenitor phenotype.

“2. The coherence between the FACS sorting strategy and the assignment to the different clusters is questionable. Indeed, many hematopoietic cells were detected in the data set while they are supposed to be EpCAM-. Furthermore, it needs to be explained why a considerable number of fetal and mature hepatocytes, that are normally EpCAM-, were identified within the EpCAM+ population scRNA-seq data.”

Reviewer 3 – Point 2: Explanation of EpCAM+ cell heterogeneity:

We thank the reviewer for these helpful observations and suggestions. To clarify:

- (i) We would firstly like to politely draw attention to the fact that in the literature it is well established EpCAM is expressed in parenchymal cells of human foetal liver and therefore not at all unexpected for foetal hepatocytes, distinct from progenitors to be isolated with the EpCAM⁺ FACS population^{2,29,30}. Indeed the observation that most foetal hepatocytes come from the EpCAM⁺/NCAM⁻ cells while HHyPs come exclusively from the EpCAM⁺/NCAM⁺ cells is, we would argue, excellent evidence our FACS strategy was effectively designed and isolated different human foetal liver hepatic populations (Fig. 1B,E).
- (ii) We agree that many hematopoietic cells were detected in the data set. The assertion however that those cells must be EpCAM⁻ has no supporting evidence from the literature as far as we are aware. Indeed this study is the first in depth single cell analysis of human foetal liver to be performed before which, no clear markers for haemopoietic cells in human foetal liver were reported. Furthermore, the phenotypic analysis of non-parenchymal cell types identified in this study clearly identified discriminant markers for example of stromal cells - *BGN* and *DCN* (Fig. S2) – supporting the conclusion that the workflow employed successfully identified unique, discrete and novel populations.
- (iii) To confirm our FACS strategy did not permit sequencing of contaminating cells, we performed FACS analysis of human foetal liver for markers identified in the haemopoietic cell population identified by expression of erythrocyte markers *CINRIP1*, *KLF1* and *TFRC*. In this way, we were able to isolate cells from human foetal liver that were CD235a⁻/EpCAM⁺/TFRC⁺ confirming that non-parenchymal cell types identified were bona fide and not a technical issue with the FACS sorting strategy (Fig. S2C-D). Furthermore, this population is significantly enriched in *EpCAM* gene expression (Fig. S1D) thus unlikely to be a contaminant. It will be of great interest to further explore the phenotype and function of these cells however this is beyond the parameters of this current study.
- (iv) EpCAM⁺ adult cells expressing mature hepatocyte markers were also noted in our data set. We explore the explanation for this in depth using our pseudo-lineage analysis to identify distinct populations of EpCAM⁺ cells enriched for either mature hepatic or biliary markers. These cells likely represent transitional phenotypes between the cells the reviewer is referring to (Fig. S6).

Fig. S2C-D: Transcript expression of TFRC overlaid on the 2D t-SNE space of human foetal liver scRNA-seq analysis. TFRC is enriched in the erythro-like foetal human liver cell cluster. CD235a⁻/EpCAM⁺/TFRC⁺ cells were subsequently FACS sorted from human foetal liver cell to confirm these are not a contaminating cell type, and expected to be included in our strategy.

“3. The population of HHyPs is assumed to be similar in adult and fetal tissues, however, their transcriptomic profiles seem to indicate that they represent two distinct populations. A more in-depth analysis of their shared (or distinct) characteristics is therefore crucial.”

Reviewer 3 – Point 3: Shared and different characteristics between adult and foetal HHyPs:

We thank the reviewer for raising this interesting point and accept the original manuscript may have caused confusion characterising similarities and differences of foetal vs. adult HHyPs. To address this we have:

- (i) Performed a more in-depth analysis of their shared and distinct characteristics (Fig. 3B-D) utilizing several publicly available data sets of different hepatic populations to further interrogate differences between foetal HHyPs, adult HHyPs and mature BECs (Fig. 6)
7,24,25
- (ii) Identified unique gene expression signatures that define a human foetal HHyP phenotype, which we utilize to FACS isolate HHyPs for *in vivo* repopulation studies (Fig. 5).
- (iii) Revised the manuscript to address the possibility of adult HHyPs representing a sub population of mature BECs (page 15, line 7).

Fig. 3B,D: Comparison of foetal and adult HHyP significantly enriched genes. Transcript expression of selected markers overlaid on the 2D t-SNE space of human liver scRNA-seq analysis for adult HHyP restricted genes (i), foetal HHyP restricted genes (ii) and other cell type-specific expression patterns (iii) (D). We identify unique gene expression signatures that distinguish foetal and adult HHyP phenotypes.

C

Foetal HHyP enriched pathways

Fig. 3C: Gene set enrichment analysis (GSEA) of foetal vs adult HHyP for Gene ontology (GO) terms. We identify enrichment of several ‘stem-cell’ related biological processes in foetal HHyPs compared to adult HHyPs.

“4. The reliability of the results supporting the existence of a hepatobiliary hybrid progenitor population, as identified in mouse, is not clear. For example, figure 1E displays the distribution of expression of a subset of genes in HHyP versus the few cholangiocytes from the scRNA-seq data, compared to bulk results from HepPD and BilPD in mouse. The authors concluded that the HHyPs have very similar expression patterns to HepPDs. First of all, the analysis would benefit from a more global comparison and not only based on a few genes. The expression of the orthologous top marker genes of the HepPDs should be shown in the human data. Furthermore, it is not possible to directly compare $\log(\text{TPM})$ values from different data sets and even less from bulk versus single-cell RNA-seq. It would be more informative to indicate for example the fold changes and if the differences are significant. The HHyPs can also be compared against different cell types and not only cholangiocytes. The authors could also investigate the possibility to apply powerful batch correction techniques such as scmap (Kiselev et al., BioRxiv, 2017) to project the single-cell results onto the referenced cell types from the bulk RNA-seq.”

Reviewer 3 – Point 4: Comparison of foetal HHyP transcriptome with mouse oval cells and other hepatic progenitor-like cells.

We welcome the suggestions made here but would respectfully suggest that the manner in which our analysis was performed may have been misinterpreted. We completely agree for example that it is not possible to directly compare gene expression values from scRNA-seq Vs foetal analysis, therefore we looked at fold change between liver progenitor phenotypes with their respective biliary populations to identify genes significantly up or down regulated in both (Fig. S4). We have now however also expanded our analysis to include a higher number of genes to help address this confusion. As suggested by the reviewer, we also in addition now compare HHyPs against different cell types both in our study and the studies of others including Tarlow *et al* (2015)²⁵ (FIG. S4), MacParland *et al* (2018)²⁴ (Fig. S3) and Fu *et al* (2019)⁷ (Fig. 3E). These results are summarized in figure 6.

Fig. S3: Violin plots of marker genes in HHyp and mature cholangiocyte clusters alongside Bulk RNA-seq gene expression data for mouse hepatocyte-derived proliferative ducts (HepPD) and biliary-derived proliferative ducts (BiPD)²⁵. We identify a signature of genes enriched in both mouse and human bipotent progenitor phenotypes vs biliary committed cells including genes *SFRP5*, *CAV1* and *MCAM*.

Figure 6

Fig .6: Heatmaps of selected marker expression in populations identified in this study with different cohorts of liver progenitor-like cells, biliary epithelial cells (BECs) and primary hepatocytes. Shown is gene expression patterns of publicly available sequencing data from GEO (<https://www.ncbi.nlm.nih.gov/geo/>) or literature mined. Bulk RNA-seq data sets include reprogrammed human hepatic liver progenitor cells (hepLPCs-Heps, GSE105019) and primary hepatocytes (pHs, GSE105019) from Fu *et al* (2018)⁷ and mouse hepatocyte-derived proliferative ducts (HepPD, Tarlow *et al* 2015) and biliary-derived proliferative ducts (BilPD, Tarlow *et al* 2015)²⁵. ScRNA-seq data sets include human BECs (GSE115469) from Macparland *et al* (2018)²⁴. Expression graded as high (red) to low (blue) relative to individual data sets.

“5. While mature cholangiocytes are ALB⁺, this marker can still be expressed in biliary progenitor cells. One is therefore left to wonder what exactly the data is that excludes that HHyPs are not mere biliary progenitors? In line, AFP, which is expressed in hepatoblasts and embryonic hepatic cells, is not expressed in any fetal HHyPs. This points to a similarity between HHyPs and mouse “oval cells”, being also AFP⁻. The latter cells were long believed to be bi-potent progenitors but clear evidence for their capacity to produce mature hepatocytes is still lacking (reviewed in Kopp et al. *Nature Cell Biology*, 2016). In addition, TROP-2 was previously found to be distinctive for mouse “oval cells”, which differentiated these cells from cholangiocytes (Okabe et al., *Development*, 2009 (doi: 10.1242/dev.031369).

Taken together, it would be valuable with respect to understanding the true nature of HHyPs if the authors could perform a comparison of transcription profiles between HHyPs and rodent oval cells.”

Reviewer 3 – Point 5: Similarity between HHyPs and mouse “oval cells”.

We identify that HepPDs isolated after mouse chronic liver injury provide the most accurate gene expression data set to represent mouse “oval cells”²⁵. As stated in the previous comment we perform in depth comparative study between HHyPs and ‘oval-like’ cells as reviewer 3 has requested (Fig S4). We refer to Okabe et al., *Development*, 2009, in our manuscript (page 10 line 14). While studies such as Okabe’s suggest TROP-2 is only expressed during injury, both our study and MacParland *et al* 2018 identify TROP2 has being expressed in mature BECs as well (Fig. S6E), supporting our findings that it marks a mature HHyP phenotype²⁴. It also suggests key differences between mouse and human liver progenitor biology, highlighting the importance of our study using human data.

Fig. S6E: *In situ* staining of TROP-2 (red) single-plex and in duplex with STAT1 (green) in the bile ducts (BD) of uninjured human adult liver.

“6. The authors used 3D organoids to validate the functional relevance of EpCAM+ sorted cells. To prove the functional proximity of cultured cells used for in-vitro differentiation to HHyPs, it is necessary to provide a comparative analysis (transcriptomics). In addition, it would be valuable if the authors would support their IF data with gene expression profiling to highlight that TROP-2 negative cells are capable of giving rise to both cell types.”

Reviewer 3 – Point 6: Justification of ex-vivo organoid culture experiments

We would politely direct the reviewer to our responses to reviewers 1 and 2 on the same topic - 3D organoid culture in this study was not used for functional analysis. We are confident that our *in vivo* analysis FACS isolating HHyPs including on TROP2⁺ expression as requested by reviewer 3 addresses concerns regarding HHyP lineage potential. Furthermore we have included new *in vivo* data to validate the differentiation potential of TROP-2 negative foetal human liver HHyPs.

“7. The evidence supporting the bipotency of HHyPs is, as already indicated, rather scarce. Indeed, the conclusions are mainly based on the pseudo-time trajectory analysis. However, the results of the latter analysis are not linked to the mature hepatocytes and cholangiocytes questioning the biological relevance of the branching event. How would this trajectory look like if mature cells would have been included? One can also wonder why no fetal cells are located along the two branches as duct cells and hepatocytes are also present in fetal livers, questioning once again the biological relevance of this analysis as well as the similarity between adult and fetal HHyPs. Thus, the putative bipotency in fetal liver remains questionable. An analysis with Monocle including mature cells and/or only on adult HHyPs could potentially clarify these questions. The results of StemID are also not described well enough to be informative for the reader.”

Reviewer 3 - Point 7: Validating the bi-potency of human foetal liver HHyPs

We thank the reviewer for these valuable comments which we address as follows:

- (i) Our prediction is that no foetal cells are located along the branches of the pseudo lineage trajectory since it accurately plots a foetal to adult transition. One would expect foetal liver cells to be developmentally earlier than adult cells. From our analysis comparing HHyPs to other data sets it appears that the transition of adult hepatocytes to a bi-potent progenitor both during injury and *in vitro* reprogramming is accompanied by gene expression changes that partially re-capture a foetal HHyP phenotype identified in our study here (Fig. 3E). We have therefore updated the manuscript to reflect that pseudo-lineage may be bi-directional (Fig. S6A).
- (ii) We now provide new *in vivo* evidence to support HHyP bi-potency.
- (iii) We have removed StemID analysis as we agree with reviewers 3 comments that it is not informative enough to be useful.

Fig. 3E: Heatmap of top foetal HHyP up regulated genes in publicly available sequencing data from human hepatic liver progenitor cells (hepLPCs-Heps, GSE105019) converted from primary hepatocytes (Fu *et al* 2018)⁷. Heatmap shows expression in Primary Hepatocytes (pH) and human primary hepatocytes converted into liver progenitor-like cells (HepLPCs) at different stages in transition and expansion medium (TEM). transcriptome of a recent study for human primary hepatocyte-derived liver progenitor-like cells.

- (iv) We identified that many of the top genes enriched in foetal HHyPs are enriched during the transition process from primary hepatocytes to progenitor-like cells including MCAM, ANXA2, ANXA4, BICC1, SPIN1, TNFRSF12A, STAT1 and ABCC3. This suggests that *in vitro* reprogramming techniques to create progenitor-like cells from mature hepatocytes are moving towards a foetal HHyP-like phenotype.

“8. The fetal and adult HHyPs seem to constitute distinct populations and the authors don't provide sufficient evidence supporting their similarity (see also point 3). Moreover, TROP-2 expression, identified as a key marker to differentiate cells committed toward the biliary lineage from uncommitted HHyPs, seems to also separate fetal (TROP-2-) and adult cells HHyPs (TROP-2+). It is therefore questionable whether the lineage tracing analysis of mixed HHyPs (fetal and adult) is unbiased. Specifically, TROP-2 was identified as a ductal marker in the pseudo state 3. Since this population consists of only adult cells, it might highlight a developmental stage rather than a specific cellular commitment state. The analysis might be more convincing if performed separately for both fetal and adult HHyPs and if involving more cells. It would thereby be equally valuable if independent analytical validation would be sought for the trajectory analysis using for example the RNA velocity tool (La Manno et al., Nature, 2018).”

Reviewer 3 – Point 8: Distinction between foetal and adult HHyPs

We thank the reviewer for these valuable comments. In our manuscript we have provided global transcriptional comparison of foetal and adult HHyPs to identify similarities and differences, and how this relates to mature hepatic and biliary populations. *TROP-2* expression is actually identified due to its specific expression in adult HHyPs and BECs compared to foetal HHyPs. During pseudo-lineage analysis we identify a potential biliary lineage committed population based on enrichment of markers including *TROP-2*, along with *KRT23* and *CXCL8* amongst others. *TROP-2* positive cells, coming exclusively from adult are unlikely to represent a developmental stage as *TROP-2* expression is found in adult human liver bile ducts and isolated here from uninjured adult liver. It is more likely to define either a transitional phenotype reflecting liver cell plasticity or a defined biliary sub-population. We have revised our discussion to better address these points (page 16, line 8).

It has recently been reported that adult mice bile ducts contain distinct cholangiocyte populations with differing proliferative capabilities³¹. Our findings suggest that transcriptionally distinct adult biliary populations are present in healthy liver. It is possible that these populations and the mechanisms that govern their behaviour regulate the inherent plasticity of parenchymal cell populations towards mature hepatic and biliary lineage in response to injury^{6,25,32-35}.

“9. The single-cell RNA-seq analysis workflow description is rather shallow, rendering it difficult to evaluate its quality / robustness.”

Reviewer 3 – Point 9: Single-cell RNA-seq analysis workflow description

We thank the reviewer for raising this point, unfortunately however it is not clear what further information is being requested. The manuscript outlines in detail the FACS isolation (page 17, line 4), DNA library preparation (page 17, line 16 – line 23), pre and post alignment protocol including relevant computational programs (page 18, line 1 – line 5). We also include supplementary information on the quality control strategy to remove ‘bad’ cells (Fig. S1B-C), which represents a well-established and published pipeline for scRNA-seq analysis³⁶.

“Other comments:

1. The potential cholangiocytes are not indicated on the t-SNE map.”

→ This is now indicated on the t-SNE map

“2. The number of cells for each cluster is not stated.”

→ This is now stated in the Figure legend of Fig. 1

“3. The genes on the heat map are often not indicated.”

→ We have now corrected this to include gene names in heatmaps

“4. The authors do not investigate at all pseudo state 5.”

→ We have now investigated pseudo state 5 and found no gene expression patterns that inform of a specific phenotype or potential function.

References

- 1 Rashid, T., Kobayashi, T. & Nakauchi, H. Revisiting the flight of Icarus: making human organs from PSCs with large animal chimeras. *Cell Stem Cell* **15**, 406-409, doi:10.1016/j.stem.2014.09.013 (2014).
- 2 Schmelzer, E. *et al.* Human hepatic stem cells from fetal and postnatal donors. *J Exp Med* **204**, 1973-1987, doi:10.1084/jem.20061603 (2007).
- 3 Turner, R. *et al.* Human hepatic stem cell and maturational liver lineage biology. *Hepatology* **53**, 1035-1045, doi:10.1002/hep.24157 (2011).
- 4 Huch, M. *et al.* Long-term culture of genome-stable bipotent stem cells from adult human liver. *Cell* **160**, 299-312, doi:10.1016/j.cell.2014.11.050 (2015).
- 5 Nowak, G. *et al.* Identification of expandable human hepatic progenitors which differentiate into mature hepatic cells in vivo. *Gut* **54**, 972-979, doi:10.1136/gut.2005.064477 (2005).
- 6 Kim, Y. *et al.* Small molecule-mediated reprogramming of human hepatocytes into bipotent progenitor cells. *J Hepatol* **70**, 97-107, doi:10.1016/j.jhep.2018.09.007 (2019).
- 7 Fu, G. B. *et al.* Expansion and differentiation of human hepatocyte-derived liver progenitor-like cells and their use for the study of hepatotropic pathogens. *Cell Res* **29**, 8-22, doi:10.1038/s41422-018-0103-x (2019).
- 8 Seo, M. J., Suh, S. Y., Bae, Y. C. & Jung, J. S. Differentiation of human adipose stromal cells into hepatic lineage in vitro and in vivo. *Biochem Biophys Res Commun* **328**, 258-264, doi:10.1016/j.bbrc.2004.12.158 (2005).
- 9 Dan, Y. Y. *et al.* Isolation of multipotent progenitor cells from human fetal liver capable of differentiating into liver and mesenchymal lineages. *Proc Natl Acad Sci U S A* **103**, 9912-9917, doi:10.1073/pnas.0603824103 (2006).
- 10 Liu, H., Kim, Y., Sharkis, S., Marchionni, L. & Jang, Y. Y. In vivo liver regeneration potential of human induced pluripotent stem cells from diverse origins. *Sci Transl Med* **3**, 82ra39, doi:10.1126/scitranslmed.3002376 (2011).
- 11 Zhao, T. *et al.* Humanized Mice Reveal Differential Immunogenicity of Cells Derived from Autologous Induced Pluripotent Stem Cells. *Cell Stem Cell* **17**, 353-359, doi:10.1016/j.stem.2015.07.021 (2015).
- 12 Okamoto, R. *et al.* Human iPS Cell-based Liver-like Tissue Engineering at Extrahepatic Sites in Mice as a New Cell Therapy for Hemophilia B. *Cell Transplant* **27**, 299-309, doi:10.1177/0963689717751734 (2018).
- 13 Lui, K. O. *et al.* Tolerance induction to human stem cell transplants with extension to their differentiated progeny. *Nat Commun* **5**, 5629, doi:10.1038/ncomms6629 (2014).
- 14 Hannan, N. R. *et al.* Generation of multipotent foregut stem cells from human pluripotent stem cells. *Stem Cell Reports* **1**, 293-306, doi:10.1016/j.stemcr.2013.09.003 (2013).
- 15 Boyd, A. S. & Wood, K. J. Characteristics of the early immune response following transplantation of mouse ES cell derived insulin-producing cell clusters. *PLoS One* **5**, e10965, doi:10.1371/journal.pone.0010965 (2010).
- 16 Jung, K. B. *et al.* and. *FASEB J* **32**, 111-122, doi:10.1096/fj.201700504R (2018).
- 17 Sambathkumar, R. *et al.* Generation of hepatocyte- and endocrine pancreatic-like cells from human induced endodermal progenitor cells. *PLoS One* **13**, e0197046, doi:10.1371/journal.pone.0197046 (2018).
- 18 Cruz-Acuña, R. *et al.* Synthetic hydrogels for human intestinal organoid generation and colonic wound repair. *Nat Cell Biol* **19**, 1326-1335, doi:10.1038/ncb3632 (2017).
- 19 Figliuzzi, M. *et al.* Bone marrow-derived mesenchymal stem cells improve islet graft function in diabetic rats. *Transplant Proc* **41**, 1797-1800, doi:10.1016/j.transproceed.2008.11.015 (2009).
- 20 Leong, K. G., Wang, B. E., Johnson, L. & Gao, W. Q. Generation of a prostate from a single adult stem cell. *Nature* **456**, 804-808, doi:10.1038/nature07427 (2008).
- 21 Chan, C. K. *et al.* Identification and specification of the mouse skeletal stem cell. *Cell* **160**, 285-298, doi:10.1016/j.cell.2014.12.002 (2015).
- 22 Yang, L. *et al.* A single-cell transcriptomic analysis reveals precise pathways and regulatory mechanisms underlying hepatoblast differentiation. *Hepatology* **66**, 1387-1401, doi:10.1002/hep.29353 (2017).
- 23 Halpern, K. B. *et al.* Erratum: Single-cell spatial reconstruction reveals global division of labour in the mammalian liver. *Nature* **543**, 742, doi:10.1038/nature21729 (2017).
- 24 MacParland, S. A. *et al.* Single cell RNA sequencing of human liver reveals distinct intrahepatic macrophage populations. *Nat Commun* **9**, 4383, doi:10.1038/s41467-018-06318-7 (2018).

- 25 Tarlow, B. D. *et al.* Bipotential adult liver progenitors are derived from chronically injured mature hepatocytes. *Cell Stem Cell* **15**, 605-618, doi:10.1016/j.stem.2014.09.008 (2014).
- 26 Schmelzer, E., Wauthier, E. & Reid, L. M. The phenotypes of pluripotent human hepatic progenitors. *Stem Cells* **24**, 1852-1858, doi:10.1634/stemcells.2006-0036 (2006).
- 27 Lu, W. Y. *et al.* Hepatic progenitor cells of biliary origin with liver repopulation capacity. *Nat Cell Biol* **17**, 971-983, doi:10.1038/ncb3203 (2015).
- 28 Camp, J. G. *et al.* Multilineage communication regulates human liver bud development from pluripotency. *Nature* **546**, 533-538, doi:10.1038/nature22796 (2017).
- 29 Dollé, L. *et al.* EpCAM and the biology of hepatic stem/progenitor cells. *Am J Physiol Gastrointest Liver Physiol* **308**, G233-250, doi:10.1152/ajpgi.00069.2014 (2015).
- 30 Gires, O. EpCAM in hepatocytes and their progenitors. *J Hepatol* **56**, 490-492, doi:10.1016/j.jhep.2011.05.036 (2012).
- 31 Li, B. *et al.* Adult Mouse Liver Contains Two Distinct Populations of Cholangiocytes. *Stem Cell Reports* **9**, 478-489, doi:10.1016/j.stemcr.2017.06.003 (2017).
- 32 Schaub, J. R. *et al.* De novo formation of the biliary system by TGF β -mediated hepatocyte transdifferentiation. *Nature* **557**, 247-251, doi:10.1038/s41586-018-0075-5 (2018).
- 33 Raven, A. *et al.* Cholangiocytes act as facultative liver stem cells during impaired hepatocyte regeneration. *Nature* **547**, 350-354, doi:10.1038/nature23015 (2017).
- 34 Tanimizu, N. *et al.* Progressive induction of hepatocyte progenitor cells in chronically injured liver. *Sci Rep* **7**, 39990, doi:10.1038/srep39990 (2017).
- 35 Katsuda, T. *et al.* Conversion of Terminally Committed Hepatocytes to Culturable Bipotent Progenitor Cells with Regenerative Capacity. *Cell Stem Cell* **20**, 41-55, doi:10.1016/j.stem.2016.10.007 (2017).
- 36 McCarthy, D. J., Campbell, K. R., Lun, A. T. & Wills, Q. F. Scater: pre-processing, quality control, normalization and visualization of single-cell RNA-seq data in R. *Bioinformatics* **33**, 1179-1186, doi:10.1093/bioinformatics/btw777 (2017).

Reviewers' Comments:

Reviewer #1:

Remarks to the Author:

The cell transplant experiments under the renal capsule are not functional experiments that explore the functional stem cell capacity of the isolated cells -as was requested.

In other words do the transplanted cells have clonality, bipotentiality and can relevant show functional rescue. I acknowledge that the authors don't want to change the cells phenotype through in vitro culture prior to transplantation.

I think the authors should explain in a clear manner what these renal capsule transplant experiments show. There is clearly much good data in the manuscript but I think acknowledging the limitations of the transplant data would be constructive to the community and perhaps inspire others to follow this work with functional experiments.

Reviewer #2:

Remarks to the Author:

NCOMMS-18-21213

Single cell analysis of human foetal liver captures the transcriptional profile of an anatomically restricted hepatobiliary hybrid progenitor population

Segal JM, et al.

The single cell RNA sequencing (scRNA-Seq) analysis of human liver cell populations, comparing those from both fetal and adult stages, is quite informative and of considerable use and interest to the field of liver study. In the revise manuscript, the authors have properly addressed most of the concerns raised by this reviewer as well as by the others, which has significantly improved the overall quality of the manuscript. The newly added data showing the repopulation and differentiation potentials of FACS-sorted fetal liver hepatobiliary hybrid progenitor (HHyP) cells is certainly a plus to support the authors' claim on their "progenitor-like" characteristics. Even though this type of transplantation experiments do not necessarily assure that those cells indeed behave as genuine progenitor cells in situ under unperturbed conditions, it is also true that this should have been the only and practical way to evaluate the progenitor activity with human samples.

The revision of the manuscript has now made it easily understandable that fetal and adult HHyP cells exhibit clearly distinct phenotypes. This has instead raised a major concern with regard to the authors' interpretation and assignment of the adult liver HHyP cells. Obviously, the ALB+ / KRT19+ / SOX9+ HHyP cells comprised the majority of the adult liver EPCAM+ cells, while the assigned "bile duct epithelial cells (BECs)" was an extremely rare population, being located subsidiary to the major "Adult HHyP" EpCAM+ population in the tSNE plot (Figure1, B and E). This seems quite unnatural, and rather unacceptable, given the notion that EPCAM is a highly robust marker to characterize and isolate BECs in the adult liver. A more simple and reasonable interpretation is that those "adult HHyP cells" rather correspond to the authentic BEC population. That is, contrary to common belief that ALB is a hepatocyte-specific marker in the adult liver, ALB transcripts are likely expressed also in adult BECs at a certain level, thereby making those cells appear as if they are "hybrid" cells. Indeed, another scRNA-Seq data by MacParland et al. (which is referred to many times in the manuscript, as Ref. #23) has clearly indicated that BECs/Cholangiocytes express ALB at a significant level. The notion that the adult "HHyP cells" actually correspond to BECs is fully consistent with and can well explain the authors' finding that expression of TACSTD2/TROP-2 was highly restricted in the adult "HHyP cells", but not found in fetal HHyP cells, and was characteristically up-regulated during biliary lineage commitment from the fetal HHyP cells in the ductal plate to committed BECs in bile ducts in fetal livers (Figure 4D). It can also solve a problem with an unconvincing and poor-grounded argument that the authors'

“isolation and FACS strategy has captured only a small number of BECs” (page 6, line 1). The authors claim that mature “BECs” can be defined as EPCAM+ cells co-expressing trefoil factors (TFFs). It has been reported by immunohistochemical analyses that TFF1 and TFF3 were specifically expressed in large/hilar bile ducts and TFF2 in extramural peribiliary glands around large bile ducts, while that none of them was observed in small bile ducts (which include septal and interlobular bile ducts and terminal bile ductules) in normal human liver tissues. It should thus be plausible to speculate that those TFF+ “BECs” described in the manuscript actually represents a specific and minor sub-population among EPCAM+ BECs that resides in large bile ducts and the accompanying peribiliary glands (Sasaki M. et al., *Peptides* 25:763, 2004, PMID 15177870; and Sasaki M. et al., *Liver International* 24:29, 2004, PMID 15101998). Moreover, the scRNA-Seq study by MacParland et al. (Ref. #23) again has also shown that TFF2 is expressed only in a subset of EPCAM+ cholangiocytes in their liver cell samples. This notion can be easily verified by the authors by examining spatial expression profiles of transcripts for TFFs and other potential “BEC” markers.

The authors should consider these issues and further revise the manuscript accordingly. It should be noted that such a revision does not essentially affect or weaken the authors’ claim that they identified and characterized an anatomically restricted hepatobiliary hybrid progenitor population in human fetal livers.

Reviewer #3:

Remarks to the Author:

The authors have invested significant efforts to address the reviewers’ concerns and especially the in vivo validation experiment has elevated the overall quality and robustness of the paper.

Nevertheless, some concerns remain, the principal being the use of the “bipotency” term for HHyPs. Based on the revision, only fetal HHyPs are shown to possess this capacity, even though the authors often use the bi-potency term without specifying the type of HHyPs.

Indeed, the evidence for adult HHyPs to be bipotent is still scant:

1) Why did the authors perform the pseudotime trajectory analysis without including hepatocytes and cholangiocytes along with adult and foetal HHyPs, as suggested? Since Monocle will generate a pseudotime trajectory even if it doesn’t represent the evolution of the cells along a biological process, it is important to strongly validate the output. If the mature cells will be at the extremity of their supposed respective branch, this would provide at least some clue regarding the differentiation potential of adult HHyPs. Please note however that Monocle considers only unidirectional trajectories, so de-differentiation events would be poorly captured by this analytical approach. As indicated in the first review, a valuable alternative to Monocle would be the velocity tool (La Manno et al., *Nature*, 2018) to gain more insights regarding the direction of the trajectory.

2) Although the difference between BECs and adult HHyPs is clearer now, the authors are still unable to provide convincing arguments supporting the distinction of adult HHyPs from “oval cells”. The transcriptional signature of HepPD cells seems very similar to adult HHyPs (Fig S4), although, surprisingly, TROP2 expression is not shown. Both “oval cells” and adult HHyPs are TROP2+ (in contrast to fetal HHyPs) and reside in perioral regions. The in vivo experiments supporting the bipotency of HHyPs were performed on TROP2- cells, i.e. fetal. It remains therefore unclear why the authors insist on terming adult cells that are clearly reminiscent of “oval cells”, HHyP-like?

Other potential issues:

1) The authors mention that they “looked at fold change between liver progenitor phenotypes with their respective biliary populations to identify genes significantly up or down regulated in both (Fig. S4)”. Since the actual values to compare are not the log(TPM) values but the fold changes, it might be misleading that the actual axis is log(TPM) and that the fold changes are not directly

represented.

2) Can the authors explain why a key hepatic TF HNF4A is not detected at the RNA level in hepatocytes and HHyPs (Fig 1 and Fig2), even though its expression is detected in IF staining (Fig 5F)?

3) For the scRNA-seq analysis, the following details are still missing: which criteria were used to filter the cells and what were the thresholds? How were the features to perform the analysis such as the tSNE and clustering selected? Which is the Fold Change cutoff to select differentially expressed genes? Etc.

4) Fig 5 F,E,G annotations are misplaced.

REVIEWERS' COMMENTS:

Reviewer #1 (Remarks to the Author):

The cell transplant experiments under the renal capsule are not functional experiments that explore the functional stem cell capacity of the isolated cells -as was requested. In other words do the transplanted cells have clonality, bipotentiality and can relevant show functional rescue. I acknowledge that the authors don't want to change the cells phenotype through in vitro culture prior to transplantation.

I think the authors should explain in a clear manner what these renal capsule transplant experiments show. There is clearly much good data in the manuscript but I think acknowledging the limitations of the transplant data would be constructive to the community and perhaps inspire others to follow this work with functional experiments.

Response

We are very grateful for this positive feedback. We also however acknowledge the comments that the renal capsule experiments do not provide evidence of clonality or functional rescue and therefore agree with reviewer 1 that the cells of interest (HHyPs) do not meet criteria for “stem-ness”. We strongly believe though that the combination of *in situ* (Fig. 4A-D, Fig. S3), *ex vivo* (Fig 4E-F) and *in vivo* (Fig. 5, Fig. S7) data support our hypothesis that foetal HHyPs have proliferative capacity with hybrid hepato-biliary lineage potential - therefore satisfying criteria for them to be labelled “hybrid progenitors”. *For exactly these reasons, we have been very careful to avoid labelling our population of interest as “stem cells” but instead chose the term “progenitors”.*

Regarding the selection of experimental approach; as per reviewer 2’s comments; “... **this should have been the only and practical way to evaluate the progenitor activity with human samples**” and reviewer 1’s acknowledgement “**I acknowledge that the authors don't want to change the cells phenotype through in vitro culture prior to transplantation**” we feel our decision to validate functional capacity in vivo using the experimental approach chosen was vindicated.

Revisions

- We have revised the manuscript to reflect the limitations of our study, in particular the conclusions that can be drawn from our renal capsule experiments. In our discussion we have provided clarification on the reasoning behind our experimental approach (**page 16, line 10:17**), and the limitations we and the wider field face (**page 15, line 19:25**).
- We have clarified that our approach is not a true functional validation of ‘stemness’ properties of our HHyP populations regarding clonality and bi-potency and explain why we unable to perform functional rescue experiments that would allow this without *ex vivo* expansion (**page 16, line 4:10, line 17:20**)
- Further to this we have tailored our conclusions that can be drawn from renal capsule transplantation experiments (**page 16, line 17:23**).

- We suggest how future experiments may address these limitations (page 17, line 15:18).

Reviewer #2 (Remarks to the Author):

NCOMMS-18-21213

Single cell analysis of human foetal liver captures the transcriptional profile of an anatomically restricted hepatobiliary hybrid progenitor population
Segal JM, et al.

The single cell RNA sequencing (scRNA-Seq) analysis of human liver cell populations, comparing those from both fetal and adult stages, is quite informative and of considerable use and interest to the field of liver study. In the revise manuscript, the authors have properly addressed most of the concerns raised by this reviewer as well as by the others, which has significantly improved the overall quality of the manuscript. The newly added data showing the repopulation and differentiation potentials of FACS-sorted fetal liver hepatobiliary hybrid progenitor (HHyP) cells is certainly a plus to support the authors' claim on their "progenitor-like" characteristics. Even though this type of transplantation experiments do not necessarily assure that those cells indeed behave as genuine progenitor cells in situ under unperturbed conditions, it is also true that this should have been the only and practical way to evaluate the progenitor activity with human samples.

Response

We appreciate reviewer 2's very positive comments regarding the importance of our study for the field. We also agree that given the context of our study which used freshly isolated human foetal and adult samples, the renal capsule transplantation model was the only practical way to evaluate HHyP populations in vivo.

The revision of the manuscript has now made it easily understandable that fetal and adult HHyP cells exhibit clearly distinct phenotypes. This has instead raised a major concern with regard to the authors' interpretation and assignment of the adult liver HHyP cells. Obviously, the ALB+ / KRT19+ / SOX9+ HHyP cells comprised the majority of the adult liver EPCAM+ cells, while the assigned "bile duct epithelial cells (BECs)" was an extremely rare population, being located subsidiary to the major "Adult HHyP" EpCAM+ population in the tSNE plot (Figure1, B and E). This seems quite unnatural, and rather unacceptable, given the notion that EPCAM is a highly robust marker to characterize and isolate BECs in the adult liver. A more simple and reasonable interpretation is that those "adult HHyP cells" rather correspond to the authentic BEC population. That is, contrary to common belief that ALB is a hepatocyte-specific marker in the adult liver, ALB transcripts are likely expressed also in adult BECs at a certain level, thereby making those cells appear as if they are "hybrid" cells. Indeed, another scRNA-Seq data by MacParland et al. (which is referred to many times in the manuscript, as Ref. #23) has clearly indicated that BECs/Cholangiocytes express ALB at a significant level. The notion that the adult "HHyP cells" actually correspond to BECs is fully consistent with and can well explain the authors' finding that expression of TACSTD2/TROP-2 was highly restricted in the adult "HHyP cells", but not found in fetal HHyP cells, and was characteristically up-regulated during biliary lineage commitment from the fetal HHyP cells in the ductal plate to committed BECs in bile ducts in fetal livers (Figure 4D). It can also solve a problem with an unconvincing and poor-grounded

argument that the authors’ “isolation and FACS strategy has captured only a small number of BECs” (page 6, line 1).

Response

We are glad the revised manuscript has been accepted to provide clearer definition between foetal and adult HHyP populations. We also accept that further clarification may be required regarding the origin, phenotype and function of our ‘adult HHyP’ population. While adult HHyPs (as they are currently labeled) express many markers of mature BECs (*KRT7*, *KRT19*, *SOX9*), we were surprised to identify a large number of EpCAM⁺ cells as being highly *ALB*⁺ and expressing other hepatic markers including *TF*, *APOA2* and *APOC3* (Fig1. D). Having identified a sub-population of adult HHyPs negative for ALB, we labelled these as mature BECs based on traditional marker expression patterns found in the literature.

We also accept the quite interesting hypothesis that adult HHyPs may indeed represent a sub-population of mature BECs distinct from ALB⁻/TFF1⁺/TFF2⁺/TFF3⁺ cells. Our heatmap analysis also confirms that adult HHyPs and mature BECs (as labelled in our study) have distinct gene expression profiles (Fig. 6). However, our opinion is that we cannot say conclusively that these cells are a specific mature BEC population. Nor does the data confirm that the ALB⁺ BECs identified in McParland *et al* (2018)¹ are not potentially an adult human liver progenitor-like population, since no functional validation has been performed on those cells by anyone in the field to date (at least to the best of our knowledge). Without the possibility of human lineage tracing experiments, we can therefore speculate these cells to potentially be one of the following:

- 1) A population of TROP2⁺ human HHyPs maintained in uninjured adult liver, that retain transcriptional phenotype of foetal HHyPs
- 2) A TROP2⁺/ALB⁺ sub-population of mature BECs that reside in intra-hepatic bile ducts.
- 3) A TROP2⁺/ALB⁺ hepatic cell restricted to the limiting plate that maintains biliary marker expression.

Our transcriptomic comparison with the mature BEC population from MacParland *et al* (2018)¹ strongly validates the distinct phenotypes of adult biliary populations identified in our study. We briefly discuss that our study suggests “that transcriptionally distinct adult biliary populations are present in healthy liver” (page 16, line 7) which fits with recent findings in the literature from Li *et al* (Ref#43)² and as reviewer 2 mentions below in the Sasaki M. *et al* papers^{3,4}.

Revisions

We have revised our manuscript to discuss the potential of adult HHyPs being a sub-population of BECs and the implications this could have.

- We have expanded our discussion to include the possibility that adult HHyPs actually represent a distinct population of BECs. We draw on the findings of MacParland *et al* (2018) who also find *ALB*⁺ cells in their mature BEC population¹ (**page 15, line 1:10, page 15 line 16:23**).
- We discuss how as reviewer 2 commented, TROP-2 expression being absent in foetal HHyPs and present in adult HHyPs and BECs is consistent with this notion (**page 15, line 10:19**).
- We have removed the statement “isolation and FACS strategy has captured only a small number of BECs” on **page 6, line 1** (as reviewer 2 has suggested).

The authors claim that mature “BECs” can be defined as EPCAM⁺ cells co-expressing trefoil factors (TFFs). It has been reported by immunohistochemical analyses that TFF1 and TFF3

were specifically expressed in large/hilar bile ducts and TFF2 in extramural peribiliary glands around large bile ducts, while that none of them was observed in small bile ducts (which include septal and interlobular bile ducts and terminal bile ductules) in normal human liver tissues. It should thus be plausible to speculate that those TFF+ “BECs” described in the manuscript actually represents a specific and minor sub-population among EPCAM+ BECs that resides in large bile ducts and the accompanying peribiliary glands (Sasaki M. et al., Peptides 25:763, 2004, PMID 15177870; and Sasaki M. et al., Liver International 24:29, 2004, PMID 15101998). Moreover, the scRNA-Seq study by MacParland et al. (Ref. #23) again has also shown that TFF2 is expressed only in a subset of EPCAM+ cholangiocytes in their liver cell samples. This notion can be easily verified by the authors by examining spatial expression profiles of transcripts for TFFs and other potential “BEC” markers.

Response

We appreciate reviewer 2 leading us to these research articles. We believe this further strengthens the conclusions of our study that distinct adult biliary populations were indeed identified here. While it is evident there is a strong possibility that we are capturing BEC heterogeneity in our study between the ‘adult HHyP’ and ‘mature BEC’ clusters, our validation does not rule out the possibility of a specific adult HHyP population distinct from mature BECs. In situ staining of adult non-injured liver reveals markers highly expressed in the foetal HHyP-like phenotype enriched in the bile duct and in hepatic zones surrounding the portal mesenchyme (Fig. S5D). Spatial expression profiling of transcripts for TFFs is an excellent suggestion but outside the scope of this manuscript.

Revisions

- We have included further commentary in the discussion to acknowledge the possibility that the distinct biliary populations identified in our scRNA-seq may relate to anatomically restricted sub-populations present in different intra-hepatic ductal regions. We reference Sasaki M. et al (2004) and Sasaki M. et al (2004) in this discussion point (page 15, line 3:6).

The authors should consider these issues and further revise the manuscript accordingly. It should be noted that such a revision does not essentially affect or weaken the authors’ claim that they identified and characterized an anatomically restricted hepatobiliary hybrid progenitor population in human fetal livers.

We agree that uncovering new complexity of BEC heterogeneity would not weaken the claim of our study and is in fact an area we plan to explore further in follow up studies. We would therefore revise the manuscript in the discussion to reflect the possibility of this. However, as reviewer 3 discusses below, adult HHyPs display a similar expression pattern to mouse ‘oval cells’, with clear distinction between BECs and adult HHyPs. We believe it is important to maintain this distinction but include further discussion to incorporate all potential interpretations from our data set.

--

Reviewer #3(Remarks To The Authors):

The authors have invested significant efforts to address the reviewers’ concerns and especially

the *in vivo* validation experiment has elevated the overall quality and robustness of the paper. Nevertheless, some concerns remain, the principal being the use of the “bipotency” term for HHyPs. Based on the revision, only fetal HHyPs are shown to possess this capacity, even though the authors often use the bi-potency term without specifying the type of HHyPs.

Indeed, the evidence for adult HHyPs to be bipotent is still scant:

Response

We appreciate reviewer 3’s comments regarding the improvement in the quality and robustness of the paper. Much focus of our revisions was on the functional validation of foetal HHyPs. We accept that within the manuscript we do not consistently clarify which HHyP population is being referred to specifically.

Revisions

We have revised the manuscript to

- Specifically clarify which HHyP population we are referring to throughout the text (**throughout manuscript**).
- We make it clear that *in vivo* validation and therefore any conclusions drawn from this only apply to foetal HHyPs. Our expanded discussion regarding the potential of adult HHyPs reflects this (**page 16, line 4:25 and page 17, line 13:15**).
- We have also revised sub headings to provide more distinction between different HHyP populations being investigated.

Why did the authors perform the pseudotime trajectory analysis without including hepatocytes and cholangiocytes along with adult and foetal HHyPs, as suggested? Since Monocle will generate a pseudotime trajectory even if it doesn’t represent the evolution of the cells along a biological process, it is important to strongly validate the output. If the mature cells will be at the extremity of their supposed respective branch, this would provide at least some clue regarding the differentiation potential of adult HHyPs. Please note however that Monocle considers only unidirectional trajectories, so de-differentiation events would be poorly captured by this analytical approach. As indicated in the first review, a valuable alternative to Monocle would be the velocity tool (La Manno et al., Nature, 2018) to gain more insights regarding the direction of the trajectory.

Response

Our original submission of the manuscript did include pseudo-lineage analysis containing adult hepatocyte populations. This was removed in the revised version as we felt no additional information was included that could not be delineated from comparing foetal and adult hepatic populations. In the original comments, reviewer 3 rightly pointed out that pseudo-lineage analysis seemed to separate foetal and adult HHyPs by *TROP2* expression (Reviewer 3, point 8). Furthermore, upon request from reviewer 2 in the original comments we revised our pseudo-lineage plot to reflect the potential bi-directionality of lineage relationships between progenitor, hepatic and biliary populations that may occur in adult injury settings as well as development (Reviewer 2, point 2).

Monocle will always generate a pseudo time trajectory regardless of whether it reflects a true biological event, thus we were cautious about including further analysis. We have run multiple pseudo-lineage analyses using foetal and mature populations, and did not identify results we

feel were worthy of inclusion in this study. Technical variation still has powerful influence on single cell analysis. This makes comparing adult and foetal cells through computational tools such as Monocle challenging, as the technical differences between the isolation protocols of these different tissue sources can influence the output. We therefore had higher confidence in directly comparing foetal and adult populations, which we have validated (e.g expression of TROP-2) using *in situ* staining (Fig 4d), *ex-vivo* organoid (Fig. 4e-f) culture and *in vivo* transplantation (Fig.5).

We agree with reviewer 3 that pseudo-lineage tools such as Monocle and Velocity could provide valuable information regarding the lineage relationship between different cell types. However, the power of our data set lies in being able to isolate potentially rare defined sub-populations in both foetal and adult human liver for comparison. Potentially if we had used a less defined sort strategy with greater cell numbers, pseudo lineage would provide a more powerful tool for us to delineate lineage trajectories, especially with the rare foetal populations. However, these may not even have been captured using this approach. In reviewer 3's original comments they suggest pseudo lineage analysis would be more convincing with more cells. Therefore, we feel more analysis of this type is unlikely to add further value to the results of this study.

Although the difference between BECs and adult HHyPs is clearer now, the authors are still unable to provide convincing arguments supporting the distinction of adult HHyPs from “oval cells”. The transcriptional signature of HepPD cells seems very similar to adult HHyPs (Fig S4), although, surprisingly, TROP2 expression is not shown. Both “oval cells” and adult HHyPs are TROP2+ (in contrast to fetal HHyPs) and reside in perioral regions. The *in vivo* experiments supporting the bipotency of HHyPs were performed on TROP2- cells, i.e. fetal. It remains therefore unclear why the authors insist on terming adult cells that are clearly reminiscent of “oval cells”, HHyP-like?

Response

We agree with reviewer 3 that *TROP2* expression should be shown in Fig. S4, as it is a key marker in delineating between HepPDs/HHyPs and BilPDs/BECs⁵. (This data is included in the heatmap analysis in Figure 6, which shows that *TROP2* is not expressed in foetal HHyPs, or murine HepPDs, but is expressed in adult HHyPs, human BECs and murine BilPDs Fig. 6)⁵. For the transcriptional comparison of our data with HepPD/BilPD mouse cells from Tarlow *et al* (2014)⁵, both foetal and adult HHyPs are clustered together, which may explain why it appears adult HHyPs and mouse ‘oval’ cells have similar transcriptional signatures. We can clarify this in the manuscript text. It may also be clearer if we use only the foetal HHyP transcriptional signature to compare with Tarlow *et al* (2014) gene expression data⁵, in line with our response to reviewer 2's comments to delineate further the potential phenotype of adult HHyPs. It is also important to clarify that mouse oval cells have been suggested to be *TROP2*⁺, while BECs are *TROP2*⁻ (6). However, in Tarlow *et al* (2014), HepPD's, which are identified as the bi-potent ‘oval-like’ cells that arise from hepatocyte dedifferentiation are *TROP2*⁻, while BilPDs which are biliary lineage specific are *TROP2*⁺⁵. This fits with our human data that *TROP2* is highly expressed in mature BECs, but not expressed in bi-potent foetal HHyPs. That *TROP2* is expressed in adult HHyPs, suggests they are more phenotypically similar to mature BECs than ‘oval-like’ *TROP2*⁻ cells.

Revisions

We have revised the manuscript to include

- Expression of TROP-2 in Supplementary Fig4. It supports our findings that TROP2 is not expressed in murine HepPDs, but is expressed in adult HHyPs, human BECs and murine BilPDs (**Supplementary Figure 4**).
- We have expanded the textual discussion (as per our reply to reviewer 2) to encompass all possible conclusions one might draw from the adult HHyP / BEC clustering (**page 15**).

Other potential issues:

1) The authors mention that they “looked at fold change between liver progenitor phenotypes with their respective biliary populations to identify genes significantly up or down regulated in both (Fig. S4)”. Since the actual values to compare are not the log(TPM) values but the fold changes, it might be misleading that the actual axis is log(TPM) and that the fold changes are not directly represented.

Response

For Fig. S4, we took the raw count data from both our data set and Tarlow *et al* (2014) and performed normalization using transcripts per million (TPM)⁵. The log₁₀(TPM) is plotted along the y axis (Fig. S4). While we do not directly compare the two data sets as one is scRNA-seq (ours) and the other is bulk (Tarlow), it is the log₁₀(TPM) that is being plotted, therefore the expression values are accurate. We agree with Reviewer 3 however, that by not including the actual fold change between the respective progenitor (HHyP/HepPD) and biliary populations (BEC/BilPD) it may not be clear what is being represented when comparing the two data sets.

Revisions

- We have revised supplementary figure 4. We have changed the y axis label to expression to avoid confusion (**supplementary figure 4**).
- We have revised supplementary figure 4 to include the actual fold change value between HHyPs and BECs in our data, and between HepPDs and BilPDs in Tarlow *et al* (2014)⁵ (**supplementary figure 4**).

2) Can the authors explain why a key hepatic TF HNF4A is not detected at the RNA level in hepatocytes and HHyPs (Fig 1 and Fig2), even though its expression is detected in IF staining (Fig 5F)?

Response

It is likely that our read depth was too low to accurately detect *HNF4A* transcripts. This can be a common issue for transcription factors with low mRNA counts in scRNA-seq analysis. This can be seen in supplementary figure 1, where HNF4A transcript expression is detected at low amounts even in mature hepatocytes (Fig. S1D). *In situ* mRNA staining for HNF4A using RNAScope (ACDbio), reveals high expression in both hepatocytes and HHyPs in foetal human liver (Fig. S5C), which is a much more sensitive detection method for RNA expression. In Fig 6, HNF4A protein is detected by IF, which, as with RNAScope is much easier to pick up. These cells have also been expanded in the renal capsule, therefore HNF4A expression patterns may be changed from the freshly isolated cells used in our scRNA-seq analysis.

Revisions

- We have included a line of text in our discussion to clarify that *in situ* staining confirms expression of *HNF4A* in foetal HHyPs (**page 14, line 9:10**).

3) For the scRNA-seq analysis, the following details are still missing: which criteria were used to filter the cells and what were the thresholds? How were the features to perform the analysis such as the tSNE and clustering selected? Which is the Fold Change cutoff to select differentially expressed genes? Etc.

Response

We are happy to include all the information reviewer 3 has requested above. We plan to revise the methods section to include the thresholds we used to filter cells and outline in detail the features used to perform t-SNE analysis and clustering. We can also include the fold change cut off used to select for differentially expressed genes in all our analyses.

Revisions

- We have included all information reviewer 3 has requested for the methods. We have explained the thresholds we used to filter cells, and outlined the features used to perform t-SNE and clustering analysis. We have also included the fold change cut offs and statistical tests used in this study (**page 19, line 9:20**).

4) Fig 5 F,E,G annotations are misplaced.

Response

We appreciate reviewer 3 pointing this out and will correct this typo.

Revisions

- This mistake has been corrected.

- 1 MacParland, S. A. *et al.* Single cell RNA sequencing of human liver reveals distinct intrahepatic macrophage populations. *Nat Commun* **9**, 4383, doi:10.1038/s41467-018-06318-7 (2018).
- 2 Li, B. *et al.* Adult Mouse Liver Contains Two Distinct Populations of Cholangiocytes. *Stem Cell Reports* **9**, 478-489, doi:10.1016/j.stemcr.2017.06.003 (2017).
- 3 Sasaki, M., Ikeda, H., Ohira, S., Ishikawa, A. & Nakanuma, Y. Expression of trefoil factor family 1, 2, and 3 peptide is augmented in hepatolithiasis. *Peptides* **25**, 763-770, doi:10.1016/j.peptides.2003.12.023 (2004).
- 4 Sasaki, M. *et al.* Site-characteristic expression and induction of trefoil factor family 1, 2 and 3 and malignant brain tumor-1 in normal and diseased intrahepatic bile ducts relates to biliary pathophysiology. *Liver Int* **24**, 29-37, doi:10.1111/j.1478-3231.2004.00883.x (2004).
- 5 Tarlow, B. D. *et al.* Bipotential adult liver progenitors are derived from chronically injured mature hepatocytes. *Cell Stem Cell* **15**, 605-618, doi:10.1016/j.stem.2014.09.008 (2014).
- 6 Okabe, M. *et al.* Potential hepatic stem cells reside in EpCAM+ cells of normal and injured mouse liver. *Development* **136**, 1951-1960, doi:10.1242/dev.031369 (2009).